# Arc Regulates Transcription of Genes for Plasticity, Excitability and Alzheimer’s Disease

**DOI:** 10.3390/biomedicines10081946

**Published:** 2022-08-11

**Authors:** How-Wing Leung, Gabriel Foo, Antonius VanDongen

**Affiliations:** 1Duke-NUS Medical School, Singapore 169857, Singapore; 2Department of Pharmacology and Cancer Biology, Duke University, Durham, NC 27710, USA

**Keywords:** Alzheimer’s disease, memory, chromatin, transcription

## Abstract

The immediate early gene Arc is a master regulator of synaptic function and a critical determinant of memory consolidation. Here, we show that Arc interacts with dynamic chromatin and closely associates with histone markers for active enhancers and transcription in cultured rat hippocampal neurons. Both these histone modifications, H3K27Ac and H3K9Ac, have recently been shown to be upregulated in late-onset Alzheimer’s disease (AD). When Arc induction by pharmacological network activation was prevented using a short hairpin RNA, the expression profile was altered for over 1900 genes, which included genes associated with synaptic function, neuronal plasticity, intrinsic excitability, and signalling pathways. Interestingly, about 100 Arc-dependent genes are associated with the pathophysiology of AD. When endogenous Arc expression was induced in HEK293T cells, the transcription of many neuronal genes was increased, suggesting that Arc can control expression in the absence of activated signalling pathways. Taken together, these data establish Arc as a master regulator of neuronal activity-dependent gene expression and suggest that it plays a significant role in the pathophysiology of AD.

## 1. Introduction

The neuronal immediate early gene, **Arc** (activity-regulated cytoskeletal associated) protein) [1,2] plays a critical role in memory consolidation [3,4,5,6]. Arc expression is rapidly and transiently induced by novel behavioural and sensory experiences [7,8,9,10,11], while its mRNA is enriched in dendrites and targeted to recently activated synapses, where it is locally translated [12,13]. Arc protein resides in excitatory synapses, where it controls α-amino-3-hydroxy-5-methyl-4-isoxazolepropionic acid (AMPA) receptor endocytosis [14], allowing it to act as a master regulator of synaptic function and plasticity [15,16] that implements homeostatic synaptic scaling at the neuronal network level [17,18,19,20,21]. While the synaptic role of Arc has been well documented, the observed failure to convert early to late long-term potentiation (LTP) in Arc knockout mice cannot be explained by an AMPA receptor endocytosis deficit [4]. This suggests that Arc may have additional functions.

Interestingly, Arc protein can also be localised in the nucleus, where it binds to a beta-spectrin IV isoform and associates with promyelocytic leukemia (PML) bodies [22,23,24,25], sites of epigenetic regulation of gene transcription [26,27]. Nuclear Arc has been reported to regulate transcription of the *GluA1 AMPA* receptor [28].

Recently, another nuclear function for Arc has been demonstrated: Arc interacts with the histone acetyl transferase Tip60 [29], a subunit of a chromatin-modifying complex [30,31,32]. Arc expression level correlates with the acetylation status of one of Tip60′s substrates: lysine 12 of histone 4 (H4K12) [29], a memory-associated histone mark that declines with age [33]. Interestingly, H4K12 acetylation is up-regulated in monocytes of Alzheimer’s disease (AD) patients [34].

These newly discovered nuclear functions may point to an epigenetic role for Arc in memory consolidation. We have therefore investigated Arc’s interaction with chromatin and its association with histone marks in cultured hippocampal and cortical neurons. Fluorescent microscopy experiments demonstrated a highly dynamic interaction between chromatin and Arc, as well as a tight association between Arc and histone marks for active enhancers and active transcription. RNA-Sequencing (RNA-Seq) experiments in which activity-dependent Arc expression was prevented using a short hairpin RNA (shRNA) showed that Arc regulates the transcription of over nineteen hundred genes controlling memory, cognition, synaptic function, neuronal plasticity, intrinsic excitability, and intracellular signalling. Interestingly, Arc also controls the expression of susceptibility genes for Alzheimer’s disease, as well as many genes implicated in the pathophysiology of this disorder. A gene ontology (GO) analysis identified downstream signalling pathways and diseases associated with the observed changes in mRNA levels, while an ingenuity pathway analysis (IPA) revealed upstream regulators predicted by the change in gene expression profile caused by Arc knockdown. Finally, we induced expression of the endogenous Arc gene in human embryonic kidney 293T (HEK-293T) cells, using CRISPR-Cas9, which resulted in the increased transcription of many neuronal genes. Taken together, our data demonstrate that Arc transcriptionally controls neuronal activity-dependent expression of many genes underlying higher brain functions and may be involved in the development of Alzheimer’s disease and other neurodegenerative disorders.

## 2. Materials and Methods

### 2.1. Animals and Chemicals

All experiments involving the use of animals were performed according to the guidelines of the Institutional Animal Care and Use Committee (IACUC). Time-mated E18 Sprague Dawley rats were sacrificed immediately after delivery to the vivarium. All chemicals were purchased from Sigma-Aldrich, St Louis, MO, USA, unless otherwise stated.

### 2.2. Culturing Hippocampal and Cortical Neurons

Hippocampi and cortices were dissected from E18 embryos of Sprague Dawley rats. Hippocampi or cortices underwent dissociation based on the protocol from the Papain Dissociation System (Worthington Biochemical Corporation, Lakewood, CA, USA). Gentle mechanical trituration was performed to ensure complete dissociation of tissues. Dissociated cells were plated on poly-D-lysine-coated dishes at a plating density of 1.5 × 10^5^/cm^2^ in neurobasal medium (Gibco, Grand Island, New York, NY, USA) supplemented with 10% (*v*/*v*) foetal-bovine serum (FBS), 1% (*v*/*v*), penicillin-streptomycin (P/S, Gibco, Grand Island, New York, NY, USA) and 2% (*v*/*v*) B27 supplement (Gibco, Grand Island, New York, NY, USA) for 2 h. FBS-containing medium was then removed and replaced with FBS-free medium, and cells were subsequently cultured with FBS-free to prevent astrocytic over-growth. Medium was changed on days in vitro (DIV) 5. Subsequently, medium was changed every three to four days. Experiments were carried out on DIV 18–22.

### 2.3. Pharmacological LTP Using 4BF

Hippocampal or cortical neuronal cultures were treated with a combination of 100 µM 4-aminopyridine (4AP), 50 µM bicuculline [35,36], and 50 µM forskolin for the times stated to induce pharmacological LTP and increase Arc expression [23,37,38]. This drug combination will henceforth be referred to as **4BF**.

### 2.4. Immunofluorescence

For immunofluorescence labelling, cells were fixed with 100% ice-cold methanol at −20 °C for 10 min. Cells were washed three times with 1× phosphate-buffered saline (PBS, in mM: 137 NaCl, 2.7 KCl, and 12 phosphate buffer) containing 0.1% (*v*/*v*) Triton X-100 (PBS-Tx). Depending on the antibodies used, some cells were fixed again with 4% (*w*/*v*) paraformaldehyde (PFA) in 1× PBS containing 4% (*w*/*v*) sucrose. Cells were washed three times in 1× PBS-Tx and blocked in 2% (*w*/*v*) bovine serum albumin (BSA) in 1× PBS for 1 h at room temperature (rtp). Depending on the species the secondary antibodies were raised in, 10% (*v*/*v*) serum of the corresponding species was added to the blocking buffer. Cells were probed with primary antibodies as indicated for the experiments: (i) anti-Arc (1:300, Santa Cruz, Dallas, TX, USA, sc-17839), (ii) anti-Arc (1:300, Synaptic Systems, Goettingen, Lower Saxony Land, Germany, 156 003), (iii) anti-MAP2 (1:300, Millipore, Temecula, CA, USA, AB5622), (iv) anti-H3K27Ac (1:300, Wako, Osaka, Japan, 306-34849) and (v) anti-H3K9Ac-S10P (1:300, Abcam, Cambridge, United Kingdom, ab12181) in antibody dilution buffer (1× PBS containing 1% (*w*/*v*) bovine serum albumin (BSA), 5% (*v*/*v*) serum and 0.05% (*v*/*v*) Triton X-100) for 1 h at rtp. Cells were washed three times in 1× PBS-Tx. Cells were then probed with 1:1000 anti-mouse secondary antibodies coupled with Alexa-Fluor 647, Alexa -Fluor 568, or Alexa-Fluor 488 (Molecular Probes, Eugene, OR, USA) for 1 h at rtp. Cells were washed three times, followed by staining of DNA with 50 µM 4′,6-diamidino-2-phenylindole (DAPI) for 20 min at rtp. Cells were mounted in FluorSave (Calbiochem, San Diego, CA, USA). For immunofluorescence staining for Stochastic Optical Reconstruction Microscopy (STORM) imaging, cells were fixed with 3% paraformaldehyde and quenched with 0.1% sodium borohydride (NaBH_4_) [24]. Blocking, primary, and secondary antibody staining were carried out as above. A post-fixation was carried out after secondary antibody binding [24].

### 2.5. Inhibition of Arc Expression by an shRNA

Four Arc shRNA plasmids (SureSilencing, Qiagen, Valencia, CA, USA) were transfected into neuronal cultures using Lipofectamine 2000 (Qiagen, Carlsbard, CA, USA). Pharmacological LTP was induced in neuronal cell cultures using a 4 h treatment with **4BF**. Cells were fixed and stained for Arc protein. Immunofluorescence images were obtained using widefield microscopy. The effectiveness of inhibition of Arc expression was based on the co-occurrence of expression of the plasmids and the absence of Arc immunofluorescence. The most effective shRNA plasmid was chosen, and adeno-associated virus AAV9 constructs harbouring an Arc shRNA and a scrambled version of this shRNA were synthesised using the annealed oligo cloning method. The oligos for the Arc shRNA were: (i) 5′-GAT CCG GAG GAG ATC ATT CAG T-3′, (ii) 5′-ATG TCT TCC TGT CAA CAT ACT GAA TGA TCT CCT CCT TTT TG-3′, (iii) 5′-AAT TCA AAA AGG AGG AGA TCA TTC AGT-3′ and (iv) 5′-ATG TTG ACA GGA AGA CAT ACT GAA TGA TCT CCT CCG-3′. The oligos for Arc scrambled shRNA were (i) 5′-GAT CCG GTA ATT TCG GAG GAT C-3′, (ii) 5′-AAG TCT TCC TGT CAA CTT GAT CCT CCG AAA TTA CCT TTT TG-3′, (iii) 5′-AAT TCA AAA AGG TAA TTT CGG AGG ATC-3′ and (iv) 5′-AAG TTG ACA GGA AGA CTT GAT CCT CCG AAA TTA CCG-3′. The ends of the annealed oligos harbour overhangs of the restriction sites for BamH1 and EcoR1. Oligos for the Arc shRNA were annealed in buffer A (mM) 100 NaCl and 50 HEPES, pH 7.4, while oligos for Arc-scrambled shRNA were annealed in buffer B (mM) 10 Tris, pH 7.5–8.0, 50 NaCl and 1 EDTA at an equimolar concentration by heating to a temperature of 95 °C for 5 min, then cooling it down to room temperature (rtp). The most optimal buffer was chosen for this annealing step. The annealed oligos were ligated using T4 ligase (New England Biolabs, Ipswich, MA, USA) into the BamH1/EcoR1-cut vector pENN.AAV.U6.shRLuc.CMV.eGFP.SV40, generously provided by the University of Pennsylvania, Vector Core. Ligated products were transformed into Stbl3 competent cells (Thermo Fisher Scientific, Waltham, MA, USA). Successful constructs were identified by restriction enzyme digestion and verified by sequencing. AAV9 viruses harbouring the transgenes (concentrations at 1 × 10^13^–1 × 10^14^ GC/mL range) were synthesised by the University of Pennsylvania, Vector Core. Arc expression was prevented by treating neuronal cultures with 3 × 10^6^ multiplicity of infection (MOI) AAV9 Arc shRNA virus on DIV14. The induction of Arc expression by pharmacological LTP (see below) was performed between DIV19 and- DIV22.

### 2.6. Transfection of Neuronal Cultures

The Arc-eYFP construct was generated as described in [22]. Neuronal cultures (DIV16) were transfected with Arc-eYFP and H2B-mCherry (Addgene, Cambridge, MA, USA, 20972) with Lipofectamine 2000 (Invitrogen, Carlsbad, CA, USA) according to the manufacturer’s protocol with some adjustment. Arc-eYFP:H2B-mCherry DNA was added to Lipofectamine at a ratio of 1:1. The Lipofectamine:DNA complex was incubated at rtp for 20 min before being added to the cells. The complex was added dropwise such that it was evenly distributed on the cell culture. Culture medium was added after 20 min and experiments were performed on DIV19.

### 2.7. Widefield Microscopy

Fluorescence images were obtained using widefield microscopy as detailed in [24]. Images obtained were analysed using NIS Elements AR version 4.1 (Nikon) to perform background subtraction. Out-of-focus fluorescence was removed using 3D deconvolution (AutoQuant, Media Cybernetics, Rockville, MD, USA). The Region-Of-Interest (ROI) analysis tool was used to mark nuclei based on DAPI intensity. The corresponding mean Arc intensity of each nucleus was also measured using the automated measurement module. The averages of the mean Arc intensity for all neurons from non-**4BF** stimulated controls were obtained for each set of experiments. This would be used as a cut-off threshold between Arc-positive and Arc-negative neurons for each set of experiments since Arc expression was only observed upon stimulation [39,40]. This “cut-off” obtained from the non-4BF of each experiment can better account for the varying intensities, especially the less-induced and lower-intensity Arc in 4BF-treated cells. Nuclei images were cropped individually and analysed using a custom MATLAB (Math-Works, Natick, MA, USA) program. Size and intensity thresholds were applied to identify and quantitate puncta in each nucleus. Batch processing using the same size and intensity threshold was performed. The mean size of the puncta and the number of puncta were recorded. ROI ID for each nucleus was used to correlate the mean Arc intensity with the mean area or number of puncta. Statistical analysis was performed using GraphPad Prism Version 6.01. (San Diego, CA, USA) Statistical data shown are mean ± S.E.M. (standard error of the mean) across experiments.

### 2.8. Spinning Disc Confocal Microscopy

Fluorescence images and time-lapse movies were obtained using a motorised Ti-E inverted microscope (Nikon) with a 60× oil Plan-Apo objective (1.49 NA) and a 100X Apo-TIRF objective (1.49 NA). Spinning disk confocal microscopy was achieved using the CSU- W1 Nipkow spinning disk confocal unit (Yokogawa Electric, Tokyo, Japan). An sCMOS camera (Zyla, Andor, Darmstadt, Hesse, Germany) was used to capture the confocal images. Laser lines used were 488 nm (100 mW) for GFP, 515 nm (100 mW) for eYFP, and 561 nm (150 mW) for mCherry (Cube lasers, Coherent, Santa Clara, CA, USA). Fast excitation/emission switching was obtained using a dichroic beam splitter (Di01-T405/488/568/647-13 × 15 × 0.5, Semrock, Rochester, New York, NY, USA) and filter wheels controlled by a MAC6000DC (Ludl, Hawthorne, New York, NY, USA). The Perfect Focus System (Nikon, Tokyo, Japan) was applied to ensure minimal focus drift during image acquisition. Z stacks were obtained using step sizes recommended for the objectives used, which were processed using 3D blind deconvolution (AutoQuant, Albany, New York, NY, USA) to remove out-of-focus fluorescence.

### 2.9. Stochastic Optical Reconstruction Microscopy (STORM)

Dual-colour STORM image sequences were obtained using a Zeiss ELYRA PS.1 platform (Oberkochen, Baden-Wurttemberg, Germany). Endogenous Arc and the dual histone marker H3K9Ac-S10P were labelled with primary antibodies and visualised using Alexa 488 and Alexa 647 secondary antibodies. Time-lapse movies of 10,000 frames were obtained of neuronal nuclei expressing Arc capturing the blinking of individual Alexa 488 and 647 molecules brought into the dark state by intense laser illumination. Fitting of a 2D Gaussian function to each blinking dot allowed their XY localisation to be determined with high precision (typically 30 nm). Super resolution images were generated from the localisations by superimposing a 2D Gaussian (green for 488 nm, red for 647 nm) for each localised position. Molecule localisation and image rendering were performed by the Zen software blue and black edition, Zeiss, Oberkochen, Baden-Wurttemberg, Germany.

### 2.10. Cell lysate Preparation and Western Blotting

Following **4BF** stimulation, neuronal cultures were washed gently with 1× PBS. Cells were gently scraped off and harvested in an Eppendorf tube. Cells were spun down at 10,000× *g* for 5 min at 4 °C to obtain the cell pellet. Total protein was isolated using an RNA-protein extraction kit (Macherey-Nagel, Düren, North Rhine–Westphalia, Germany), as specified by the manufacturer. A BCA kit (Pierce, Rockford, IL, USA) was used to measure the concentration of proteins. A total of 30 µg of each protein sample was denatured and reduced by boiling at 95 °C for 5 min in 10% (*v*/*v*) 2-mercaptoethanol-containing Laemmli sample buffer (Bio-Rad, Hercules, CA, USA). Samples were resolved by SDS-PAGE with a pre-cast Tris-glycine gel (Bio-Rad, Hercules, CA, USA) and transferred onto PVDF membranes using the Trans-Blot Turbo Transfer System (Bio-Rad, Hercules, CA, USA) as indicated by the manufacturer. Membranes were blocked for 1 h at rtp with 5% (*w*/*v*) non-fat milk block (Bio-Rad, Hercules, CA, USA) in 1× Tris buffered saline (TBS) (in mM) (140 NaCl, 3 KCl, 25 Tris base) (First Base, Singapore) containing 0.1% (*v*/*v*) Tween-20 (TBST), followed by primary antibody incubation for 1 h (anti-Arc, 1:1000, Santa Cruz, Dallas, TX, USA, sc-17839) in 1× TBST at rtp. Membranes were washed three times, each for 5 min in 1× TBST at rtp. Secondary antibody binding was performed using the corresponding HRP-conjugated secondary (1:10,000, Invitrogen, Carlsbad, CA, USA) for 1 h in 1× TBST at rtp. Protein bands were detected with chemiluminescence substrate (Pierce, Rockford, IL, USA) visualised with a Gel Doc XRS imaging system (Bio-RAD, Hercules, CA, USA) or developed on scientific imaging film (Kodak, Rochester, New York, NY, USA).

### 2.11. RNA Sample Preparation, Library Construction, RNA-Seq

**4BF-**treated neuronal cells were washed, scraped, and spun down as above. RNA samples were obtained from the cell pellet using the RNA-protein extraction kit as specified by the manufacturer (Macherey-Nagel, Düren, North Rhine–Westphalia). Library construction and RNA sequencing were performed by the Duke-NUS Genome Biology Facility. An amount of 2.2 µg of RNA was used for library construction. Prior to library construction, the quality of the RNA was analysed with an Agilent 2100 Bioanalyzer (Palo Alto, CA, USA). Following poly-A enrichment, recovered RNA was processed using the Illumina TruSeq stranded mRNA kit (San Diego, CA, USA) to generate the adaptor-ligated libraries. A total of 9 samples were analysed. These samples came from 3 different sets of experiments (*n* = 3). Each set contained samples treated with (i) 8 h **4BF**, (ii) Arc shRNA + 8 h **4BF** and (iii) Arc scrambled shRNA + 8 h **4BF**. Prior to RNA- sequencing, the samples were also on analysed with 0 h 4BF on the RT-PCR to ensure Arc induction. Six samples were sequenced per lane on the HiSeq 3000 using 150 pair-end reads. For the HEK293T cells, RNA was obtained similarly. Three samples were analysed, with two Arc-induced samples and one control sample. The samples were processed as described above and sequenced on 1 lane on the HiSeq 3000.

### 2.12. Computational Analyses of RNA-Seq Data

FASTQ files obtained from the RNA-sequencing were mapped to the rat genome using Partek Flow (version 7.0.18.1210) (Partek Inc., St. Louis, MO, USA). Adapter sequences were trimmed. Contaminant reads contributed from rDNA, tRNA and mtDNA were filtered out using Bowtie2 (version 2.2.5) (within Partek Flow, Partek Inc., St. Louis, MO, USA). Filtered, trimmed reads of high quality (Phred score > 30) were then mapped onto the *Rattus norvegicus* genome (rn6) for the rat samples or the *Homo sapiens* genome (hg38) for the HEK293T samples with Star (version 2.5.3a) (within Partek Flow, Partek Inc., St. Louis, MO, USA) [41]. Post alignment QA/QC was performed to determine if alignment had good average coverage and if reads were uniquely aligned. The unique paired reads were used for gene expression quantification. Reads were assigned to genes using the expectation/maximisation (E/M) algorithm in Partek Flow [42] based on the annotation model rn6 (Ensembl transcripts release 93) for the rat samples and the annotation model hg38 (Ensembl transcripts release 94) for the HEK293 samples. To ensure only informative genes were included in the downstream analysis, noise (maximum feature counts ≤ 30) was filtered out. Read counts between samples were normalised with the Upper Quantile method [43]. As genes with very low expression might be inadequately represented and incorrectly identified as differentially expressed, a constant of 1 was added to normalised counts for rectification. Statistical analysis was performed using the gene-specific analysis (GSA) module in Partek Flow to identify differential gene expression. *p*-value and fold changes of differentially expressed genes were calculated based on the lognormal with shrinkage distribution. Genes with an average coverage of less than 3 were also filtered out prior to statistical analysis. Differential gene expression with a cut- off value of false discovery rate (FDR) step-up < 0.05 [44] and an absolute fold change ≥ 2 was considered for further gene ontology analysis in Partek Flow, which is based on the GO Consortium (version 2018_08_01) [45,46]. Setting up cut-off at FDR step-up < 0.05 ensures statistically significant differential gene expressions were tapped with an absolute fold change of 2 to focus on those with a higher magnitude of change [47]. Coherent biological data could then be more meaningfully interpreted through gene ontology analysis [45,46]. GO analysis was also performed on the transcriptional regulators/factors that were observed to be altered upon Arc knockdown using DAVID (version 6.8) [48,49]. The EASE score obtained from the DAVID analysis is a modified Fisher Exact *p*-value to indicate gene enrichment in the annotation terms. Functional analysis on the statistically significant differential gene expression (FDR step-up < 0.05; absolute fold change ≥ 2) was performed by Ingenuity Pathway Analysis (IPA) (version 01.13, Qiagen, Redwood City, CA, USA). Pathways and their associated downstream effects, diseases, regulator networks and upstream regulators were identified by IPA. Predictions on the possible activation and inhibition of pathways, downstream effects and upstream regulators were inferred from the degree of consistency in the expression of the target genes compared to the fold changes in the differentially expressed gene list. This activation or inhibition status was expressed as a z-score, with z ≥ 2 indicating activation and z ≤ 2 indicating inhibition. Inferences made were based on at least one publication or from canonical information stored in the Ingenuity Knowledge Base. Fisher’s exact test was used to calculate the *p*-value for all analyses in IPA.

### 2.13. Plasmid Construction for Arc Expression in HEK293T Cells

The following plasmids were used: pSBbi-Hyg (Addgene #60524) and pSBbi-Pur (Addgene #60523), which were a gift from Eric Kowarz. pCMV(CAT)T7-SB100 (Addgene #34879) was a gift from Zsuzsanna Izsvak. sgRNA(MS2) cloning backbone plasmid, (Addgene #61424), MS2-P65-HSF1_GFP, (Addgene #61423), and dCAS9-VP64_GFP (Addgene #61422) were gifts from Feng Zhang. The psBbi-Hyg-dCAS9-VP64 and the pSBbi-MS2-P65- HSF1-Pur plasmids were constructed by isolating the dCAS9-VP64 and MS2-P65-HSF1 sequences via PCR from the MS2-P65-HSF1_GFP and dCAS9-VP64_GFP plasmids and annealed into the SfiI-linearised pSBbi-Hyg and pSBbi-Pur plasmids. psBbi-Hyg-dCAS9-VP64 and pSBbi-MS2-P65-HSF1-Pur were then co-transfected with pCMV(CAT)T7-SB100 into HEK293T using JetPrime (Polyplus-Transfection, Illkirch-Graffenstaden, France) according to manufacturer’s instructions. HEK293T cells with successful transposition of both genes were selected with a combination of 0.75 µg/mL puromycin (Gibco, Grand Island, New York, NY, USA) and 200 µg/Ml hygromycin B (Nacalai Tesque, Kyoto, Japan) in DMEM + 10% FBS (Gibco, Grand Island, New York, NY, USA) over several passages for a month.

### 2.14. Transfection for Endogenous Arc Overexpression and Purification of mRNA

sgRNA(MS2) back- bone plasmids containing guide RNAs complementary to human Arc promoters were transfected into the mutated HEK293T cells. A separate control well was transfected with sgRNA(MS2) backbone containing LacZ promoter sgRNAs. After 48 h, mRNA was purified using the NucleoSpin RNA kit (Macherey-Nagel, Düren, North Rhine–Westphalia, Germany) and submitted for RNA-sequencing to the Duke-NUS Genomics Core Facility.

The table below lists the guide RNAs used for inducing expression of endogenous Arc in HEK293T cells:
**Promoter****SgRNA Sequence**Human Arc (1)GGGCGCTGGCGGG-GAGCCTGHuman Arc (2)CCTCCCGTCCCTT-GCCGCCCLacZ (1)TTCCGGCTCGTATGTT-GTGTLacZ (2)GCTTTACACTTTATGCTTCC

## 3. Results

Arc is a neuronal activity-dependent immediate early gene [1,2], whose expression is induced by exposure to a novel environment or a new sensory experience [7,8,11]. Knockdown of Arc expression abrogates long-term memory without affecting short-term memory, indicating a critical role for Arc in memory consolidation [3,4,5,6]. Arc protein localises to dendritic spines, where it regulates AMPA receptor endocytosis [14], and to the nucleus [22,28,50,51], where its function is less understood. In this study, we have used cultured hippocampal and cortical neurons to study the role of Arc in the nucleus. Arc expression can be induced by increasing network activity in neuronal cultures, using a combination of 4-aminopyridine (4AP), bicuculline, and forskolin (**4BF**), a form of pharmacological LTP [23,24,37,38]. Figure 1 shows that this form of network activation strongly induces the expression of Arc in a subset of neurons. In this in vitro paradigm, Arc localises predominantly to the nucleus four hours after network-activity-dependent induction of its expression.

Memory consolidation requires de novo gene expression [52,53], which is induced by activation of signalling cascades that originate in the synaptic connections potentiated during learning [54,55,56,57,58]. This synapse-to-nucleus signalling results in post-translational modifications of chromatin, including acetylation, methylation, phosphorylation, and sumoylation of histones and methylation of DNA [59,60]. Chromatin modification alters its nanostructure, which controls the accessibility of gene promoters to the transcription machinery [61,62]. These synaptic activity-induced epigenetic processes can alter gene expression and have been shown to be critical for learning and memory [63,64,65,66,67,68,69,70]. We therefore characterised the structure and dynamics of chromatin in cultured hippocampal neurons, evaluated how pharmacological LTP (**4BF** treatment) affected chromatin structure, and compared chromatin properties of neurons expressing Arc protein with control neurons that do not.

### 3.1. Chromatin Reorganisation in Arc-Positive Neurons

The induction of Arc protein expression by pharmacological network activation (Figure 1) is relatively slow and reaches a maximum level between 4 and 8 h. Arc is only expressed in a subset of neurons. As shown in Figure 2, chromatin organisation is different between neurons that are positive and negative for Arc. Chromatin was visualised by labelling DNA with the fluorescent dye 4′,6-diamidino-2-phenylindole (DAPI). Whereas chromatin in Arc-negative neurons is relatively homogenous, the nuclei of Arc-positive neurons contain many bright puncta, representing chromocenters with densely packed chromatin, in which genes are likely silenced (Figure 2A,B). The puncta are interspersed with domains of highly open chromatin, which is more supportive of efficient gene transcription. The number of puncta increased from 11.1 ± 0.8 puncta in Arc- negative nuclei to 15.9 ± 0.8 puncta in Arc-positive nuclei (Figure 2C). However, the mean area of the puncta was not significantly different between Arc-positive and Arc-negative neurons (Figure 2D).

### 3.2. Arc Associates with Dynamic Chromatin

The interaction between Arc and chromatin was studied in more detail using time-lapse fluorescence microscopy of hippocampal neurons expressing Arc and histone 2B (H2B) tagged with YFP and mCherry, respectively (Figure 3). Arc was induced in 18-day in vitro (DIV18) hippocampal neurons by a 4 h treatment with **4BF**. The time-lapse movies of Arc-eYFP and H2B-mCherry revealed a highly dynamic chromatin that constantly reorganises on a time scale of seconds (Appendix A). Arc is concentrated in small puncta to which the chromatin can be seen to reach out with finger-like structures, which may represent the dynamic chromatin loops described by others [71,72,73].

### 3.3. Arc Associates with a Marker of Active Enhancers

Because Arc was shown to associate with the Tip60 substrate H4K12Ac [29], we have examined interactions of Arc with other histone modifications, by comparing Arc-positive and Arc-negative neurons following pharmacological network activation. The ‘histone code’ [74] is complex and still incompletely understood. We have therefore focused on histone modifications whose function is best studied. In our survey, we have found several histone modifications for which there was a difference in nuclear organisation between Arc positive and negative neurons, including H3K9Ac, H3K4me3, and H3K14Ac (data not shown). Figure 4 illustrates the close association between Arc and H3K27Ac, which marks active enhancers [75,76]. Arc and H3K27Ac form two separate lattice-like structures that are closely inter-connected and, in some locations, appear to overlap (yellow areas in Figure 4).

### 3.4. Arc Associates with a Marker for Active Transcription

Another histone mark that showed a strong interaction with Arc was H3K9Ac-S10P, which requires the concurrent acetylation of lysine 9 of histone H3 (H3K9Ac) and phosphorylation of the neighbouring serine 10 (S10P). This dual marker indicates genomic regions undergoing active transcription [24,77,78]. Figure 5 illustrates the close interaction between Arc and this histone mark, using Stochastic Optical Reconstruction Microscopy (STORM), a form of super-resolution microscopy with a resolution of ~30 nm [79]. Both Arc and H3K9Ac-S10P are enriched at the nuclear periphery, where reorganisation of chromatin between active and inactive transcriptional states takes place [80,81]. With the increased resolution of STORM, Arc can be seen to localise to distinct puncta. H3K9Ac-S10P forms an elaborate meshwork, as expected for chromatin, but also is enriched in puncta-like domains. Arrowheads in Figure 5A indicate the close apposition between these two sets of puncta. Close inspection of the interface between the two types of puncta revealed invasions of H3K9Ac-S10P into the Arc puncta (arrows in Figure 5B), resembling the finger-like chromatin structures seen in live cell imaging (Figure 3, Appendix A).

### 3.5. Arc Regulates Activity-Dependent Gene Transcription

Because Arc was found to associate with histone marks involved in transcription activation, we wanted to investigate whether network activity-induced Arc expression alters the gene expression profile of the neurons. Four short hairpin RNAs (shRNAs) targeting the coding region of Arc were tested for their ability to suppress Arc induction following four hours of **4BF** treatment. We selected the most effective shRNA to generate an adeno-associated AAV9 virus. Because AAV9 infection itself may alter the gene expression profile, we also generated a negative control consisting of AAV9 virus encoding a scrambled version of the Arc shRNA. We performed an RNA-Seq analysis of cortical neurons expressing either the Arc shRNA or its scrambled control. When **4BF**-mediated Arc expression was prevented using the Arc shRNA (Figure 6A), mRNA levels for more than 1900 genes were altered (Figure 6B). Many gene families were affected, including those associated with plasticity (*Jun*, *Fosb*, *Bdnf*, *Dlg4*, *Egr4*, *Npas4* and *Nr4a1*), synaptic proteins (syntaxin *Stx12* and synaptotagmin *Syt3*), and neurotransmitter receptors (NMDA, AMPA, GABA, glycine, serotonin, and metabotropic glutamate receptors) (Figure 6B). Arc also regulated the expression of genes controlling intrinsic excitability: 62 genes encoding ion channels (20 K^+^, 4 Na^+^, and 9 Ca^2+^ channel subunits, 7 transient receptor potential (Trp) channels, 14 ligand-gated ion channels, 7 regulatory subunits and 1 non-selective cation channel), and 139 genes encoding transporters/pumps (for glutamate, GABA, serotonin, ADP, ATP, phosphate, glucose, inositol, alanine, cysteine, glutamine, glycine, proline, Na^+^, Ca^2+^, Cl^−^, H^+^ and Zn^2+^). These results suggest that Arc regulates activity-dependent gene expression relevant for synaptic function, neuronal plasticity, and intrinsic excitability.

Figure 7 shows the 30 top-ranking genes sorted by absolute fold change (FC) caused by the shRNA- mediated knockdown of Arc expression. Gene names are shown together with a description of their function, their fold change, false discovery rate (FDR), and references to relevant papers. Many of the top-regulated genes are involved in synapse modulation, neurotransmission, neurogenesis and neurological disorders. Interestingly, 9 out of the top 30 genes have been implicated in the pathophysiology of AD (*Fgf1*, *Slc30a4*, *Npas4*, *Cxcl1*, *Jdp2*, *Nts*, *Mmp10*, *Orai2* and *Tomm34*), while an additional 5 genes are linked to amyloid beta (Aβ) metabolism (*Mmp13*, *Mmp12*, *Slc2a13*, *Igf1r* and *Apba1*).

### 3.6. GO Analysis of Differentially Expressed Genes

A gene ontology (GO) analysis was performed on the RNA-Seq data, which aim to identify the biological processes and molecular functions altered by the reduction in Arc expression (Figure 8). Arc knockdown altered many genes involved in the regulation of nervous system development and neuronal differentiation (Figure 8A). In addition, many of the genes were enriched in biological processes involved in cognition, regulation of cell projection organisation and axonogenesis (Figure 8A), processes which could modulate the structural plasticity involved in neural development, learning and memory [143,144]. While the top ten regulated genes enriched for the regulation of plasma membrane bounded cell projection organisation were both up- and down-regulated (Figure 8Cii), genes enriched for cognition and the regulation of axonogenesis were mostly down-regulated due to the absence of Arc (Figure 8Ci,Ciii). Many of the altered genes were also enriched in molecular functions such as ion channel regulator activity, glutamate receptor binding and ligand-gated ion channel activity (Figure 8B), including *Sgk1* (Figure 8Di), *Dlg4*, which encodes PSD-95 (Figure 8Dii), and *Grin2c*, which encodes the NMDA receptor NR2C subunit (Figure 8Diii). These molecular functions are well-established to underlie synaptic plasticity processes crucial for formation of memory [145,146].

### 3.7. Arc Regulates Expression of Synaptic and Plasticity Genes

The GO results in Figure 8 indicated that the knockdown of Arc affected many genes involved in synaptic plasticity, as well as genes implicated in processes underlying learning and memory. We have therefore investigated how Arc knockdown affected genes encoding synaptic proteins by manually curating a list of differentially expressed genes whose protein products are located at the presynaptic or postsynaptic compartment. A total of 232 synaptic genes were differentially expressed. *Ephb3*, *Lrfn2*, *Lama5*, *Neurod2*, *Sema4f*, *Caprin2*, and *Unc5c* are involved in the development and growth of axons and dendrites, while *Npas4, Pcdh8, Ephb3, Lrfn2, Bdnf, Atxn1, Cbln2, Cadps2, Caprin2, C1ql1, C1ql3*, and *Unc5c*, modulate the function of synapses and dendritic spines (Figure 9).

Many of these synaptic genes are also involved in neuroplasticity, cognition, learning and memory, including *Syt3, Pcdh8, Pdyn, Lrfn2, Dlg4, Kcna4, Bdnf and Mapki8ip2.*
Figure 10 lists neuroplasticity genes and genes that are involved in cognition, learning and memory, whose activity-dependent expression is regulated by Arc. Most of these genes were downregulated when activity-dependent Arc expression was prevented.

### 3.8. Arc Knockdown Altered Synaptogenesis, Synaptic Plasticity and Neuroinflammation Pathways

From the GO results and the list of manually curated synaptic genes, we were interested in investigating the signalling pathways and the possible downstream effects resulting from Arc knockdown. We have analysed the differentially expressed genes and their respective fold changes using IPA. Figure 11A shows the top 15 pathways that were altered due to Arc knockdown. IPA made inferences on the activation or inhibition of the pathways based on the differential expression observed and canonical information stored in the Ingenuity Knowledge Base. The degree of activation or inhibition of each identified pathway is indicated by the z-score. The ratio is calculated as the number of differentially expressed genes for each pathway divided by the total number of genes involved in that pathway. Many identified pathways involved cellular signalling cascades, including those mediated by CDK5, PTEN, integrin and corticotropin-releasing hormone (Figure 11A). Pathways predicted to be responsible for the observed differential expression profile include opioid and endocannabinoid signalling, synaptogenesis, synaptic long-term depression (LTD) and neuroinflammation (Figure 11A,B). *Kcnj5*, *Ptgs2*, *Grin2c*, *Cacng4* and *Gnaq* are members of at least two of the pathways shown and are synaptic genes or associated with cognition (Figure 8, Figure 9, Figure 10 and Figure 11A). Except for the neuroinflammation signalling pathway, all these pathways are associated with synaptic plasticity. Knockdown of Arc-modulated neurotransmission, synaptic plasticity, spine formation/maintenance and neurite outgrowth are processes that are crucial for learning and memory (Figure 11B) [202,203,204]. Interestingly, the two hallmarks of AD, the generation, clearance, and accumulation of amyloid beta (Aβ) and the formation of neurofibrillary tangles (NFTs), are both affected by downregulation of the neuroinflammation signalling pathway resulting from Arc knockdown (Figure 11B). These alterations in the generation and clearance of molecular markers and triggers of AD could indicate a possible role of Arc in the pathophysiology of AD [205].

### 3.9. Arc Knockdown Changes the Expression of Alzheimer’s Disease Genes

Considering that the generation, clearance and accumulation of amyloid beta and neurofibrillary tangles was predicted to be altered due to the knockdown of Arc, we investigated whether any neurological diseases or psychological disorders were correlated with the profile of differentially expressed genes mediated by Arc knock-down. Figure 12 summarises the disease annotation and predicted activation state for two disease/disorder classes whose associated genes were significantly altered by Arc knockdown.

Absence of Arc was predicted to increase damage of the cerebral cortex and its neurons and cells. In addition, Arc knockdown was also associated with psychological disorders, including Huntington’s disease, basal ganglia disorder, central nervous system (CNS) amyloidosis, tauopathy and Alzheimer’s disease. Of note, CNS amyloidosis and tauopathy are predictors of AD. The activation states of the five psychological disorders were not reported, possibly due to inconsistencies in the literature findings with respect to fold changes of the differentially expressed genes. However, the *p*-values for all five disorders were highly significant, suggesting that the progression of these disorders may be modulated by Arc function.

We next investigated how Arc knockdown could affect genes that were previously identified to increase susceptibility to AD. We have manually curated genes that were found to be genetic risk factors of AD and validated them by referencing the genome-wide association studies (GWAS) catalogue [206]. Notably, critical genetic risk factors of AD such as *Picalm*, *Apoe*, *Slc24a4*, and *Clu* were downregulated upon the knockdown of Arc [207,208,209,210,211] (Figure 13), indicating that activity-induced Arc expression is linked to enhanced transcription of these genes. Out of a total of 39 AD susceptibility genes identified, 26 were regulated by Arc (Figure 13).

Because Arc plays a role in the aetiology of AD by modulating its genetic risk factors, we investigated whether Arc regulates genes that are more broadly involved in the pathophysiology of AD. Figure 14 lists the results. While some differentially expressed genes control amyloid beta formation/accumulation through the regulation of cleavage and stabilisation of amyloid precursor protein (APP) (*Mmp13*, *Slc2a13*, *Apba1*, *Casp8, Ptgs2*, *Gpr3, Pawr, Timp3, Kcnip3, Plk2, Aplp2, Bace2, Apoe* and *Apba2*), others are involved in the hyperphosphorylation of tau and formation of neurofibrillary tangles (*Npas4*, *Cxcl1*, *Dryrk2*, *Tril*, *Pltp, Plk2* and *Selenop*). Arc knockdown also altered the expression of genes that are associated with the neurodegeneration and neurotoxicity observed in AD (*Casp8*, *Bcl2l11*, *Alg2*, *Tac1, Bdnf, Hmox1, Pawr, Ccl2, Selenop* and *Atf6*). Finally, Arc regulated genes associated with altered cognitive function, a characteristic of AD (*Mmp13, Pdyn, Tac1*, *Bdnf*, *Nr4a2*, *Penk*, *Pltp* and *Ccl2*). To date, presenilin 1 (*Psen1*) and glycogen synthase kinase 3 beta (*Gsk3b*) are the only AD mediators that have been reported to physically associate and interact with Arc [212,213,214]. Arc also interacts with endophilin 2/3 and dynamin and recruits them to early/recycling endosomes to traffic APP and beta secretase 1 (BACE1), crucial determinants of AD progression [214]. However, the observation that knocking down Arc resulted in more than 100 differentially expressed genes that are either AD susceptibility genes or genes implicated in the pathophysiology of AD (Figure 13 and Figure 14) suggests that Arc could be mediating the expression of these genes via transcriptional regulation and not simply physical interactions. Arc has previously been reported to reside in the nucleus [22,28,50,51], and we have shown how Arc physically associates with chromatin and with markers of active transcription and enhancers (Figure 3, Figure 4 and Figure 5). Therefore, we wanted to investigate how Arc downregulation affects transcription regulation.

### 3.10. Arc Regulates the Expression of Transcription Factors

From our GO analysis and a manual curation based on literature citations, we have identified 369 transcriptional regulators and transcription factors whose expression is controlled by Arc. Figure 15 shows the top 40 transcriptional regulators or factors whose mRNA levels were altered when activity-dependent Arc expression was prevented. Some of the transcriptional regulators are involved in neuronal development and differentiation (*Fgf1*, *Tgfb1i1, Fezf2*, *Jun*, *Magel2, Neurod2, Atxn1, Gdf15, Prdm1, Mycn, Nr4a2 and Pou2f2*), while others are involved in the development of neurological or neurodegenerative diseases (*Npas4, Igf1r*, *Txnip*, *Lgr4*, *Cebpd*, *Pim1, Magel2, Ireb2, Smad7, Sorbs1, Nfil3, Pknox2, Hdac9, Hmox1, Atxn1, Cbfb, Lrp2, Hipk3* and *Nr4a2*). Many of the transcriptional regulators/factors have been implicated in memory formation and plasticity, such as *Thbs1*, *Jun, Tet3, Fosb*, *Atxn1* and *Cbfb*. A GO analysis by DAVID [49] was carried out to identify the biological processes that these transcription factors could be modulating. Figure 16 shows the top 20 biological processes that were regulated by altered transcription factor expression and that have neurological relevance. Corroborating the identified functions of the top 40 transcriptional regulators/factors (Figure 15), differentially expressed transcriptional regulators/factors were observed to be highly enriched in biological processes such as differentiation of neurons, nervous system development, learning, long-term memory and aging (Figure 16). Some of the transcriptional regulators were involved in multiple processes: *Npas4*, *Jun*, *Bdnf*, *Nr4a2* and *Elavl4* modulate learning, long-term memory, aging, neuron differentiation and nervous system development (Figure 16).

### 3.11. Upstream Regulators Associated with Arc-Dependent Genes

Because Arc knockdown resulted in the differential expression of 1945 genes (Figure 6), altering downstream pathways (Figure 11) possibly leading to disease states (Figure 12), we identified the upstream modulators that could explain the vast differential expression pattern observed. From the IPA analysis, 11 upstream regulators were predicted to critically contribute to the differential expression profile (Figure 17).

Except for *Sox2*, none of these upstream regulators were transcriptionally affected by Arc knockdown, suggesting that Arc controls their function through a different mechanism. SOX2 and HDAC4 were both activated by the absence of Arc, while the function of the remaining nine regulators was inhibited. Of note, the predicted inhibition of CREB1 (z-score = −3.5) and APP (z-score = −2.8) explains the differential expression of 100 and 94 genes, respectively (Figure 17). The 11 upstream regulators predicted by IPA control the expression of *Nr4a2*, *Slc6a1* and *Igf1r,* genes that are also involved in AD progression, neuroinflammation pathways and synaptic LTD (Figure 11A and Figure 14). We have investigated the mechanisms by which Arc could alter the function of the identified upstream regulators, resulting in the alteration of downstream pathways and AD progression. The downstream pathways investigated are (i) opioid signalling, (ii) synaptogenesis, (iii) the endocannabinoid neuronal synapse pathway, (iv) synaptic LTD and (v) neuroinflammation (Figure 18). These are also the pathways whose downstream effects we focused on in Figure 11.

APP, CREB1 and TNF are three upstream regulators identified by IPA that controlled the highest number of genes involved in the downstream pathways highlighted (Figure 18). The top five genes regulated by APP were *Igf1r* (synaptic LTD) [354], *Ptgs2* (endocannabinoid neuronal synapse pathway; neuroinflammation, AD progression) [227,355,356,357], *Jun* (neuroinflammation, AD progression) [358,359,360,361], *Dlg4* (PSD95, synaptogenesis) [362,363] and *Syn2* (synaptogenesis) [364] (Figure 18). In addition to *Ptgs2* and *Syn2*, CREB1 regulated the differential expression of *Slc6a1* (neuroinflammation) [365], *Pdyn* (opioid signalling) [366] and *Fosb* (opioid signalling) [367] (Figure 18). Interestingly, TNF, whose transcription was not altered upon knockdown of Arc, regulates 15 genes (Figure 17), the top five of which are *Casp8* (neuroinflammation) [368], *Ptgs2* (also regulated by APP and CREB1), *Gabrg2* (neuroinflammation) [369,370], *Bdnf* (synaptogenesis, neuroinflammation, AD progression) [371,372,373,374] and *Penk* (opioid signalling, AD progression) [375]. While the top CREB1-regulated genes are mainly associated with the opioid signalling pathway, APP and TNF are implicated in neuroinflammation. Triggering of the neuroinflammation pathway leads to the altered expression of AD-associated genes such as *Ptgs2*, *Jun, Bdnf, Hmox1* and *Gabbr2*.

### 3.12. Arc Over-Expression Alters Gene Expression in Human Embryonic Kidney Cells

The results presented thus far suggest that preventing Arc expression during neuronal network activation results in an altered gene expression profile affecting synaptic plasticity and cellular excitability, as well as neurodegenerative disease state. We therefore tested whether Arc could alter gene transcription outside of the context of neuronal network activation and without viral infection. We induced the expression of the endogenous Arc gene in human embryonic kidney (HEK293T) cells using a CRISPR-Cas9 approach [376] (Figure 19A). Whereas wildtype HEK293T cells expressed Arc at a very low level, targeting a transcription activator complex to its promoter increased Arc mRNA levels nearly 250-fold. This in turn altered the expression of 57 genes (absolute FC > 2, *p* < 0.05), with 54 genes upregulated and 3 genes downregulated. Many of the genes have neuronal functions (Figure 19B). We have performed a GO analysis to understand the cellular components (Figure 19C) and biological processes (Figure 19D) these differentially expressed genes were involved in. We observed many genes that are typically expressed in neurons or are synaptic components, as indicated by the following GO terms: (i) *synapse part* (*p* = 1.1 × 10^−4^), (ii) *presynapse* (*p* = 1.0 × 10^−3^), (iii) *neuron part* (*p* = 1.5 × 10^−3^) and (iv) *postsynaptic membrane* (*p* = 2.3 × 10^−3^) (Figure 19C). Differentially expressed genes upon the induction of Arc in HEK293T cells are involved in synaptic transmission processes or neuronal development, including (i) *chemical synaptic transmission* (*p* = 2.5 × 10^−4^), (ii) signal release from synapse (*p* = 1.9 × 10^−3^), (iii) interneuron precursor migration (*p* = 3.2 × 10^−3^) and (iv) *axon guidance* (*p* = 3.2 × 10^−3^) (Figure 19D). Genes that are associated with these cellular components and processes were also highly altered, including (i) *Chat* (*p* = 4.7 × 10^−85^, choline acetyltransferase) located at presynaptic terminals, synthesising acetylcholine, (ii) *Oprd1* (*p* = 2.6 × 10^−62^, δ-opioid receptor), whose activation reduces pain and improves negative emotional states, (iii) *Arx* (*p* = 1.1 × 10^−70^, Aristaless Related Homeobox), a transcription factor involved in neuronal migration and development, (iv) *Scn1b* (*p* = 6.6 × 10^−22^, Na channel β1 subunit), involved in axonal guidance, (v) *Foxa3* (*p* = 3.3 × 10^−24^, Forkhead Box A3), a transcription factor involved in the determination of neuronal fate [377,378], (vi) *Pllp* (*p* = 1.5 × 10^−25^, Plasmolipin), involved in membrane organisation and ion transport, (vii) *Slc18a3* (*p* = 1.6 × 10^−16^), a vesicular acetylcholine transporter at the presynapse, (viii) *Fndc11* (*p* = 4.4 × 10^−14^, Fibronectin Type III Domain Containing 11), a vesicular gene, and (ix) *Adgrb1* (*p* = 3.7 × 10^−12^, Adhesion G Protein-Coupled Receptor B1), localised at the postsynapse, involved in synapse organisation and cell projection morphogenesis (Figure 19B).

Together with the results obtained with Arc knockdown in neurons, this finding strongly implicates Arc as a transcriptional regulator of neuronal development, synaptic function, plasticity and intrinsic excitability.

## 4. Discussion

Activity-regulated cytoskeleton-associated protein (Arc) was discovered in 1995 as a neuronal activity-dependent immediate early gene [1,2], which is rapidly transcribed in response to network activation associated with novel experiences [7,8,9,10,11]. Knockdown of Arc expression interferes with the stabilisation of short-term memory, indicating that Arc plays a critical role in memory consolidation [3,4].

Arc’s function has been most widely studied in excitatory synapses, where it regulates the endocytosis of AMPA receptors [14,17]. Interestingly, AMPA receptor removal also underlies Aβ-induced synaptic depression and dendritic spine loss [379], processes thought to be associated with cognitive dysfunction in Alzheimer’s disease [380]. In Arc knockout mice, LTP is not stable, and dissipates within a few hours, consistent with the impaired memory consolidation observed in these mice [3,4,5,6]. However, the absence of the late form of LTP in Arc knockout mice cannot be explained by an AMPA receptor endocytosis deficit [4], indicating that Arc must have additional functions. The data presented here identify a second function for Arc: regulation of neuronal activity-dependent transcription for genes associated with synaptic plasticity, intrinsic excitability and cellular signalling. Analysis of the differentially expressed genes points to Arc’s involvement in several neurological disorders, including autism, Huntington’s disease and Alzheimer’s disease. This newly proposed role for Arc is supported by its interaction with chromatin and histone markers reported here (Figure 2, Figure 3, Figure 4 and Figure 5, Appendix A).

### 4.1. Arc and Chromatin

Pharmacological network stimulation induces Arc in a subset of cultured neurons (Figure 1). Whereas chromatin in cultured hippocampal neurons is relatively uniform, Arc-positive neurons are characterised by a larger number of densely packed heterochromatin puncta (chromocenters), likely harbouring silent genes, interspersed with highly open euchromatin domains, which are capable of active transcription (Figure 2). This result is consistent with what has been observed in vivo, where Arc-deficient mice were found to have decreased heterochromatin domains [51]. These significant changes in chromatin structure observed in Arc-positive neurons are likely associated with equally substantial alterations in gene expression profiles. The correlation between Arc expression and chromatin remodelling that we observed does not establish a causative relationship. It is possible that Arc expression requires an alteration in chromatin structure, or alternatively, Arc expression may cause chromatin remodelling. Additional experiments are needed to decide on the underlying mechanism. It is also not clear at this time what determines which neurons will express Arc following network activation, although it likely has to do with the degree of participation of individual neurons in the enhanced network activity, which in turn depends on their synaptic connectivity.

Arc appears to physically interact with DNA: time-lapse movies show dynamic chromatin loops that appear to invade Arc puncta (Figure 3, Appendix A). The interaction is transient, lasting only a few seconds. Because these Arc puncta likely contain the histone acetylase Tip60 [29], it is conceivable that this interaction alters chromatin accessibility, thereby facilitating transcription. This idea is further strengthened by the association of Arc puncta with a histone marker for active enhancers (Figure 4), as well as the close apposition between Arc puncta and puncta for a dual histone marker (H3K9Ac-S10P) that labels sites of active transcription (Figure 5). A similar result has been obtained in vivo, where cocaine administration in rats results in an increase in nuclear Arc, which then associates with H3S10P [51]. Taken together, the data presented here on the interaction of Arc and chromatin may provide a mechanism for epigenetic regulation of gene transcription as the basis for memory consolidation.

### 4.2. How Does Arc Regulate Transcription?

Preventing Arc induction during neuronal network activation affects the transcription of a very large number of genes (Figure 6). The domain structure of Arc protein appears to rule out that it can function as a transcription factor [212]. This raises the question: how does Arc regulate transcription?

One possible mechanism, discussed above, is that Arc epigenetically controls gene transcription by regulating chromatin structure (through Tip60 or other chromatin re-modellers) and modification of histones (e.g., H4K12Ac [29]). However, the differential gene expression associated with Arc knockdown is mediated through eleven upstream regulators identified by IPA (Figure 17 and Figure 18). This suggests that Arc has additional, less-direct ways of regulating transcription. Interestingly, to date, none of the eleven upstream regulator proteins have been shown to either directly interact with or be modulated by Arc. They are also not transcriptionally controlled by Arc (except for *Sox2*) (Figure 17). How, then, does Arc regulate transcription by activating or inhibiting these upstream regulators? Using IPA and its ingenuity knowledge base, we were able to identify several known *interactors* of Arc that can modulate the action of the upstream regulators, which could then subsequentially alter gene transcription (Figure 20A). Next, we will discuss the mechanisms by which four identified Arc interactors, NOTCH1, TIP60/*Kat5*, APP and GSK3B, could modulate the upstream regulators.

**NOTCH1.** NOTCH1 is a transmembrane receptor capable of signalling to the nucleus. Arc is required for the proteolytic cleavage of NOTCH1 to release its intracellular domain (NICD), which can translocate to the nucleus and alter transcription [381]. NICD regulates the expression of the transcriptional repressor BCL6 [100] and the activity of the calcium-dependent kinase CAMK4 [382], which in turn alter the localisation and the nuclear-cytoplasmic shuttling of the histone deacetylase HDAC4, thereby affecting its downstream interactions/modulation [383,384] (Figure 20B). NOTCH1 could regulate the stability, nuclear localisation and signalling of the transcription factor SOX2 through regulation of the protein kinase AKT1 and cell-surface glycoprotein CD44 [385,386,387,388] (Figure 20B). NOTCH1, through NICD, controls the expression of plasminogen activator inhibitor-1 (SERPINE1) [389], an inhibitor of thrombin (F2) [390] (Figure 20B). NOTCH1 regulates the transcriptional activity of T-cell factor 4 (TCF7L2) [391], through its interaction with the DNA-repair protein Ku70 (XRCC6) [392] (Figure 20B). NOTCH1 interacts with the nerve growth factor NR4A1/Nur77 [393], thereby modulating expression levels of the cytokine tumour necrosis factor alpha (TNF) [394]. Finally, NOTCH1 regulates the expression level of the inhibitor of apoptosis protein cIAP1/*Birc2* [395], which also affects TNF expression [396] (Figure 20B).

**TIP60/*Kat5*.** The *Kat5* gene encodes TIP60, a member of the MYST family of histone acetyl transferases, which plays important roles in chromatin remodelling and transcription regulation [397]. In the fruit fly *Drosophila,* TIP60 has been implicated in epigenetic control of learning and memory [398], while it mediates APP-induced apoptosis and lethality in a fly AD model [399]. Nuclear Arc interacts with TIP60 at perichromatin regions and recruits TIP60 to PML bodies, sites of epigenetic transcription regulation [29]. Arc levels correlate with acetylation status of H4K12, a substrate of TIP60 and a memory mark that declines with aging [33], suggesting that Arc mediates activation of TIP60. TIP60/*Kat5* facilitates the repressive action of HDAC4 through the formation of complexes with the zinc-finger transcription factor KLF4 [400,401], the cAMP-dependent transcription factor ATF3 [402,403,404] and the neurodegenerative disease protein ataxin-1 (ATXN1) [405,406] (Figure 20B). Arc’s interaction with TIP60/*Kat5* may result in a complex being formed at the cIAP1/*Birc2* promoter region [407] to mediate downstream signalling of TNF [396] (Figure 20B). TIP60/*Kat5* forms a complex with the Kaiso transcription factor ZBTB33 [408], resulting in the inhibition of the TCF7L2 transcriptional complex [409] (Figure 20B). Complexing of TIP60 with ARID1B could affect SOX2 signalling [410,411,412]. The regulation of SOX2 by TIP60/Kat5 could also have an implication on the transcriptional activity of Achaete-Scute homolog 1 (ASCL1), as SOX2 and ASCL1 regulate each other, possibly in a feedback loop [413,414].

**APP.** The functional interaction between APP and Arc is crucial for Arc’s modulation of upstream regulators (Figure 20A). Arc interacts with endophilin 2/3 (SH3GL3) and dynamin on early/recycling endosomes to alter the trafficking and localisation of APP. The association of Arc with presenilin 1 (PSEN1) promotes the trafficking of γ-secretase to endosomes and enzymatic cleavage of APP [214] (Figure 20B). The generation of amyloid beta through APP cleavage leads to altered downstream signalling, activity and production of HDAC4, SOX2 and F2 through changes in caspase-3 (CASP3) [415,416], JUN [417,418] and thrombospondin-1 (THBS1) [419,420], respectively (Figure 20B). Cleavage of APP generates a cytosolic fragment, AICD, which forms a transcriptionally active complex with TIP60 and the transcription factor FE65 [421]. AICD also modulates the ubiquitin–proteasome system (UPS) via UBE2N [422], to change downstream signalling induced by TNF [423] (Figure 20B). The modulation of the UPS via UBE2N, UBC and UBE3A [424] could implicate the ubiquitination of serum- and glucocorticoid-regulated kinase-1 (SGK-1) [425,426,427] and polyglutamine-expanded ataxin 3 (ATXN3) [428] and their ability to regulate the transcription factor cAMP responsive element binding protein 1 (CREB1) [429,430] (Figure 20B). The modulation of CREB1 would further implicate changes in expression levels of the cAMP responsive element modulator CREM [431,432,433] (Figure 20B). Finally, APP has a role in the regulation of TNF through indirect modulation of CREM [434] and direct interactions with laminin could regulate the production of TNF [435,436] (Figure 20B).

**GSK3B.** Although glycogen synthase kinase 3 beta (GSK3B) is not regulated by Arc, the promotion of cleavage of APP to amyloid beta enhances the induction and activation of GSK3B [437,438,439]. This could lead to modified downstream signalling of CREB1 [440] (Figure 20B). GSK3B is also a downstream mediator of NOTCH1 [441], PSEN1 [442], and CAMK2B [443], all of which are Arc interactors [214,381,444]. This creates an interesting situation as APP/amyloid beta is positively regulated by GSK3B [445,446], creating a positive feedback loop for amyloid beta production and its downstream signalling [437,438,439] (Figure 20B).

### 4.3. Interactions among TIP60, NOTCH1 and APP

A delicate regulatory network exists among Arc’s interactors TIP60/*Kat5*, NOTCH1 and APP (Figure 20B). Arc’s activation of the γ-secretase PSEN1 to promote cleavage of APP not only increases amyloid beta load, but also results in an increased level of the APP intracellular domain (AICD) [214,447]. AICD forms a complex with TIP60/*Kat5* to alter transcriptional activity crucial for AD progression [421,448,449,450,451,452,453] (Figure 20B). This AICD-TIP60 interaction is disrupted by NICD, formed when Arc activates NOTCH1 [381], thereby downregulating AICD signalling while promoting NICD signalling [454,455] (Figure 20B). The formation of NICD and AICD is competitive, as NOTCH1 and APP are both substrates of γ-secretase [456], whose activity is regulated by Arc [214]. In addition, the induction of TIP60 histone acetylation activity by Arc [29] could also increase the negative regulation of NOTCH1 [454] (Figure 20B). This highlights Arc as an important modulator of the relationship and downstream signalling mediated by NOTCH1, TIP60/*Kat5* and APP. Of note, the mRNA levels of *Notch1, Kat5* and *App* were not significantly altered upon knockdown of Arc, indicating that the transcriptional changes brought about were due to protein interaction and activation (Figure 20B), which is upstream of transcription (Figure 20A). However, the modulation of upstream regulators by Arc is also dependent on its subcellular localisation.

### 4.4. Arc’s Subcellular Localisation Determines Its Function

When Arc is localised outside of the nucleus, it tends to accumulate in dendrites and spines, small membrane protrusions that harbour excitatory synapses. Here, Arc controls the removal of AMPA receptors by endocytosis, allowing it to regulate synaptic efficacy [14,17]. Synaptic Arc also associates with the synaptic scaffolding protein PSD-95/*Dlg4*, which complexes with the tyrosine kinase FYN [457,458,459], allowing it to regulate brain-derived neurotrophic factor (BDNF) signalling through tyrosine receptor kinase B (TrkB), a major pathway for synapse maturation, plasticity and neurodevelopmental disorders [460]. Activation of FYN could also mediate the secretion of TNF [461] (Figure 20B). A high-affinity interaction with calcium-calmodulin kinase 2 beta (CAMK2B) targets Arc to inactive synapses, where it removes GluA1 AMPA receptors from the postsynaptic membrane surface [444].

Arc has been shown to possess both a nuclear localisation signal (NLS) and a nuclear retention domain [28], allowing it to translocate to the nucleus autonomously. Once in the nucleus, Arc has access to several other potential binding partners, including a nuclear spectrin isoform (βSpectrinIV∑5) [22] and TIP60, a subunit of a chromosome remodelling complex [29]. Association with Amida, encoded by the *Tfpt* gene (Figure 20B), facilitates Arc’s entry into the nucleus [462]. Amida is a subunit of the INO80 chromatin remodelling complex, which contains the transcriptional regulator MCRS1 [463,464]. MCRS2, an isoform of MCRS1, is associated with the MLL chromatin remodelling complex, which also contains KMT2A (MLL1) (Figure 20B). Arc’s association with Amida and possibly the INO80 and MLL complexes may provide Arc with yet another opportunity to control gene expression by altering chromatin structure.

The ability of Arc to translocate between the synapse and the nucleus, with unique functions in each subcellular compartment, further strengthens its role in memory consolidation, which requires both alterations in synaptic function and de novo gene transcription [465].

### 4.5. Arc Controls Synaptic Plasticity and Intrinsic Excitability

Arc’s well-studied ability to alter synaptic efficacy by endocytosis of AMPA receptors established it as a critical regulator of synaptic plasticity [14,17,459,466]. Whereas this mechanism of activity-dependent removal of glutamate receptors supports Arc’s role in mediating long-term depression (LTD) [467,468,469,470], it does not explain the absence of stable LTP observed in Arc knock-out mice [4]. Because late-LTP is considered a critical cellular mechanism underlying memory consolidation, the molecular and cellular mechanism by which Arc supports memory stabilisation has remained elusive. The data presented here showing that Arc transcriptionally regulates the expression of a large number of synaptic proteins, with functions in both the pre- and post-synaptic compartment (Figure 9), provide a new mechanism by which Arc can control long-lasting changes in synaptic structure and function required for memory consolidation.

Formation of a memory trace not only requires long-term changes in the strength of the synapses connecting the neurons that constitute the engram, but also stable changes in their intrinsic excitability [471,472,473]. Because Arc controls the expression of a large number of ion channels and pumps/transporters, it appears that Arc is capable of supporting this functional aspect of memory consolidation as well.

### 4.6. Arc and Alzheimer’s Disease

Alzheimer’s disease is a devastating neurodegenerative disorder [474,475] characterised by the progressive loss of both synaptic function [476] and long-term memory formation [477]. There is currently no therapy that prevents, stabilises, or reverses the progression of this disease, which is projected to take on epidemic proportions as the world population ages [478,479]. Several previous studies have revealed an association between Arc and AD. A landmark study published in 2011 showed that Arc protein is required for the formation of amyloid (Aβ) plaques [214]. Moreover, Arc protein levels are aberrantly regulated in the hippocampus of AD patients [480] and are locally upregulated around amyloid plaques [481], whereas a polymorphism in the Arc gene confers a decreased likelihood of developing AD [482]. It has been shown that spatial memory impairment is associated with dysfunctional Arc expression in the hippocampus of an AD mouse model [483].

These published results together with the data presented here suggest that aberrant expression or dysfunction of Arc contribute to the pathophysiology of AD [476,484].

### 4.7. Arc and Ad Therapy

Arc’s ability to transcriptionally regulate AD susceptibility and AD pathophysiology-related genes indicates a possibility for modifying expression and activity of Arc as a therapy for AD. Current treatments for AD are symptomatic, not effective disease-modifying cures [485,486]. Many hypotheses have been proposed to underlie the development of AD, including (i) amyloid beta aggregation, (ii) tau hyperphosphorylation, (iii) neuroinflammation, (iv) neurotransmitter dysfunction, (v) mitochondria dysfunction, (vi) glucose metabolism, (vii) vascular dysfunction and (viii) viral infection [485,487,488,489]. These hypotheses have generated many new compounds, none of which have shown efficacy in slowing cognitive decline or improving global functioning [485,488]. Arc appears to be a good therapeutic candidate for AD, because of its involvement in amyloid beta production, tau phosphorylation, neuroinflammation and neurotransmission. Moreover, we have shown that Arc can modulate the expression of many genetic risk factors and genes associated with the pathophysiology of AD (Figure 11, Figure 13, Figure 14 and Figure 19). Currently, known drugs that could increase mRNA or protein expression of Arc include antidepressant drugs [490], phencyclidine [491] and corticosterone, a memory-enhancing drug [492]. Arc expression could be altered by targeting TIP60 and PHF8, two histone modifiers that together control Arc transcription [24]. Drugs could also modulate Arc’s effect via its interactors such as TIP60 and NOTCH1. Natural and synthetic drug molecules targeting TIP60 exist, but they are currently used for cancer treatment [493]. Modulation of NOTCH1 function often involves inhibitors of γ-secretase, which would also affect APP cleavage [456,494]. These pharmaceutical modifications of Arc expression and activity could present a promising starting point for the development of a more effective AD therapy.

## 5. Conclusions

The neuronal Arc gene is critically important for the stabilisation of memories. It encodes a protein that localises to dendritic spines, where it regulates endocytosis of glutamate receptors. However, Arc can also be found in the nucleus, where its function is less understood. We find that Arc tightly associates with two markers of active DNA transcription, both of which have recently been shown to be upregulated in Alzheimer’s disease. Our results further show that Arc is a master regulator of activity-dependent gene expression, controlling mRNA levels of over 1900 genes involved in neuronal plasticity and intrinsic excitability, as well as over 100 genes implicated in the pathophysiology of AD. Because Arc function has previously been shown to be dysregulated in AD, these new findings identify Arc as new therapeutic target for the treatment of AD.

## Figures and Tables

**Figure 1 biomedicines-10-01946-f001:**
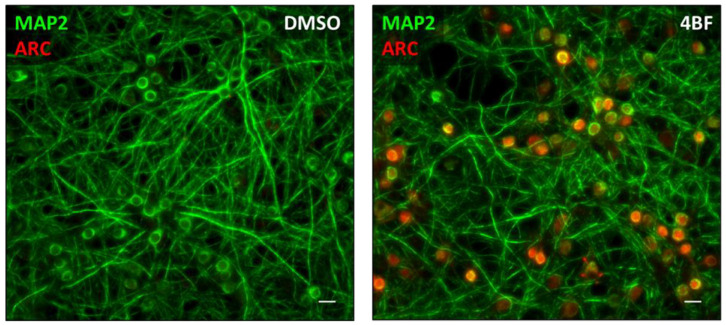
**Induction of Arc expression in hippocampal neurons by pharmacological network activation.** Hippocampal neurons (DIV19-21) were treated with **4BF**, which pharmacologically stimulates network activity and induces LTP of excitatory synapses. After 4 h of enhanced network activity, neurons were fixed and stained for Arc (red) and the neuronal marker Map2 (green). Under vehicle (DMSO) treatment, very little Arc staining could be detected (**left** panel, vehicle), whereas the increase in network activity induced strong nuclear Arc expression in approximately half of the neurons: 49 ± 8% (*n* = 3) (**right** panel, **4BF**) White scale bar is 20 µm.

**Figure 2 biomedicines-10-01946-f002:**
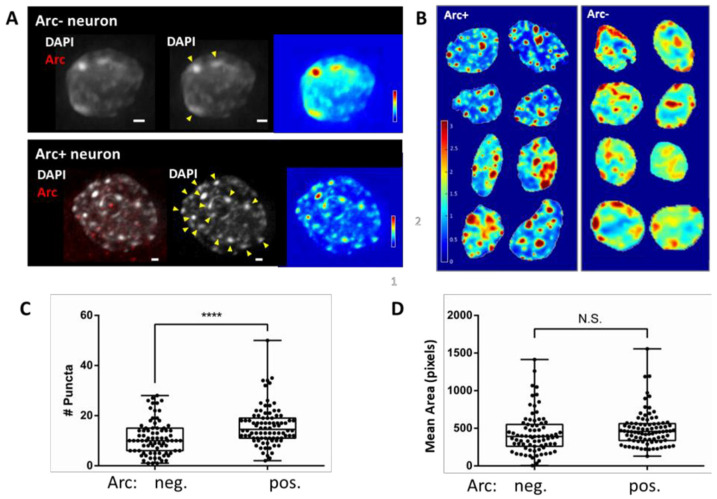
**Chromatin reorganisation in Arc-positive neurons.** Arc expression was induced in a subset of cultured hippocampal neurons by a 4 h treatment with **4BF**. Cells were fixed and stained for Arc (C7 antibody, Santa Cruz). DNA was labelled using DAPI. Z-stacks of DAPI images were obtained for neuronal nuclei that were positive and negative for Arc expression. Out-of-focus fluorescence was removed using 3D deconvolution (AutoQuant). (**A**) Max-projection images of a representative nucleus from an Arc-negative (**top**) and Arc- positive neuron (**bottom**). The white bar indicates a scale bar of 1 µm. DNA, labelled by DAPI, is shown in white while Arc expression is shown in red. Yellow arrowheads indicate DNA puncta. Heat maps of the relative DAPI intensity of the nucleus are shown in the rightmost panels. (**B**) DAPI heat maps for nuclei of 8 Arc-positive and 8 Arc-negative neurons. Relative DAPI intensity is shown by the colour scale on the left, which was the same for both panels. Whereas chromatin of Arc-negative neurons (**right** panel) was relatively homogenous (turquoise, green and yellow), Arc-positive neurons (**left** panel) were characterised by several areas of high DAPI intensity (red), indicating condensed heterochromatin (chromocenters)-separating domains with decondensed euchromatin (blue). (**C**,**D**) Puncta were quantified based on their size and intensity. Arc expression, measured as mean Arc intensity, was used to correlate with the properties of the puncta, generating the boxplots. Boxplots of number of puncta (**C**) and area of puncta (**D**) for Arc-negative and Arc-positive neurons. Each ● represents the (**C**) number or (**D**) mean area of puncta in a nucleus. A total of 167 nuclei were analysed from three sets of independent experiments. (**C**) Nuclei of Arc-positive neurons have a significantly higher number of puncta with 11.1 ± 0.8 puncta in Arc-negative nuclei and 15.9 ± 0.8 puncta in Arc-positive nuclei. **** indicates *p*-value < 0.0001, unpaired *t* test. (**D**) No significant change in area of puncta was observed: Arc-positive had an area of 488 ± 25 pixels, while the area of Arc -negative neurons was 431 ± 30 pixels (*p* = 0.15, unpaired *t* test). N.S. indicates not significant.

**Figure 3 biomedicines-10-01946-f003:**
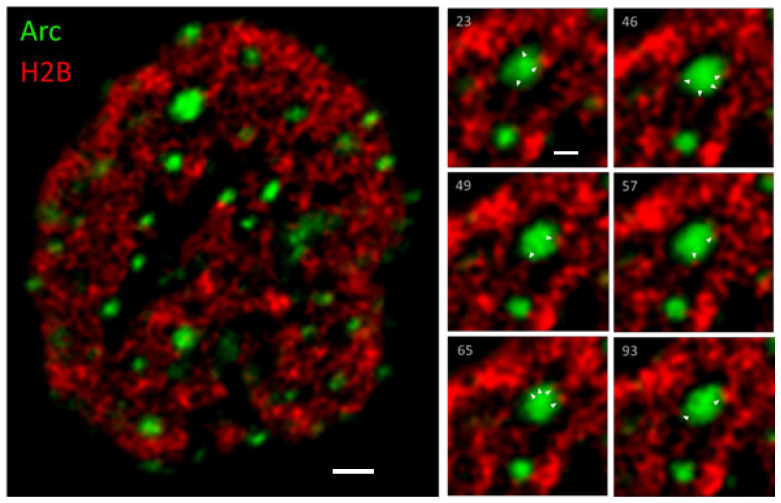
**Arc associates with dynamic chromatin.** Time-lapse movies of Arc-eYFP and H2B-mCherry expressed in hippocampal neurons (18 DIV) were obtained using a spinning disc confocal microscope (100×, 1.49 NA Apo TIRF objective). Z-stacks (5 images) were acquired for both YFP and mCherry channels. Three-dimensional blind deconvolution (AutoQuant) was used to remove out-of-focus fluorescence. The movie is 5 min long, 3.2 s between frames, which was the time required to acquire Z-stacks from both channels. The image on the left shows a single frame of the movie in the centre of the Z-stack of a neuronal nucleus (scale bar = 1 µm). Arc (green) is seen to form puncta, while H2B (red) labels the lattice-like chromatin structure. The panels on the right show six frames of a zoomed-in section illustrating small chromatin structures transiently interact with the two Arc puncta (scale bar = 500 nm). White arrowheads indicate points of contact between Arc and chromatin. The highly dynamic interaction of chromatin with Arc puncta is most clearly seen in the Appendix A.

**Figure 4 biomedicines-10-01946-f004:**
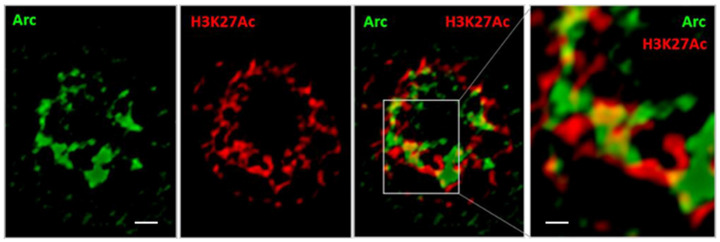
**Arc associates with H3K27Ac.** Hippocampal neurons were treated with **4BF** for 4 h, fixed with methanol and stained for Arc and H3K27Ac, which marks sites containing active enhancers. Z-stacks of images were acquired of neuronal nuclei using a spinning disc confocal microscope (60×, 1.49 NA objective). Resolution was increased using 3D blind deconvolution (AutoQuant). Scale bar is 1 µm. The enlarged section shows the close interaction between Arc and H3K27Ac. Note the elaborate interfaces between Arc (green) and H3K27Ac (red) Scale bar is 500 nm.

**Figure 5 biomedicines-10-01946-f005:**
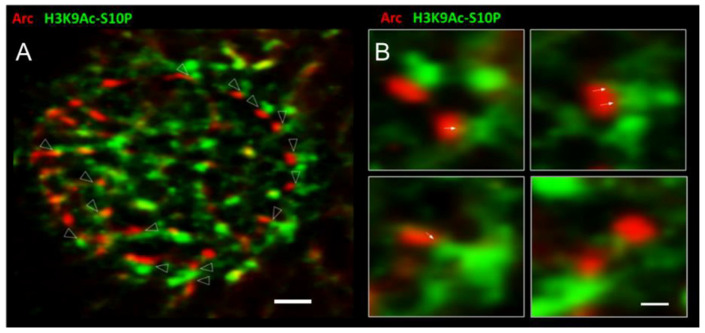
**Arc associates with H3K9Ac-S10P.** Image of a neuronal nucleus obtained using STORM. Cultured hippocampal neurons were treated with 4BF for 4 h, fixed and stained for Arc and H3K9Ac-S10P, which marks sites undergoing active transcription. (**A**) Arrowheads point to close appositions between Arc and the histone mark (scale bar = 1 µm). (**B**) Enlarged sections showing the association in greater detail. Arrows inside the Arc puncta point to what appear to be invasions of H 3K9Acc-S10P into Arc puncta (scale bar 200 nm). Regions of overlap appear in yellow.

**Figure 6 biomedicines-10-01946-f006:**
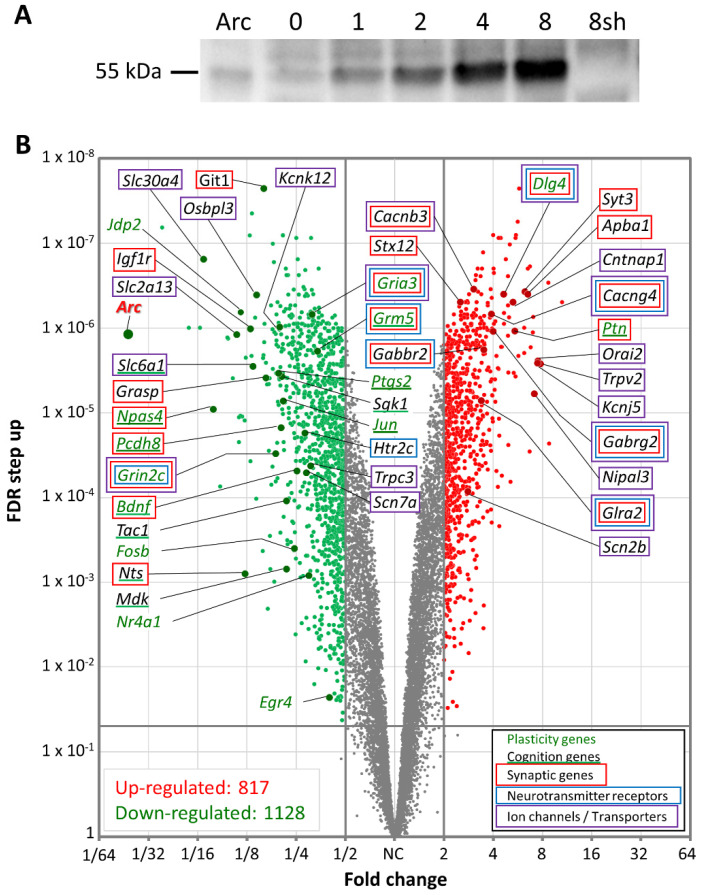
**Arc regulates gene transcription.** (**A**) Western Blot showing the time course of Arc protein expression in cultured hippocampal neurons following **4BF** treatment (time in hours indicated on top). Lane 7 (8sh) shows that Arc fails to express at 8 hours of **4BF** when the cultures are transduced with an AAV0 virus encoding a short-hairpin RNA (shRNA) targeting the coding region of Arc. Lane 1 has purified Arc protein. (**B**) volcano plot of RNA-Seq results comparing mRNA isolated from neurons after 8 hours of **4BF** that were transduced with AAV9 virus encoding either the Arc shRNA or a scrambled version of this shRNA, done in triplicated. Preventing activity-dependent Arc expression resulted in the upregulation of 817 genes (red), and down-regulation of 1128 genes (green). Genest that are below the cut-off (FDR > 0.05) or absolute fold change < 2) are marked in grey. Some of the highly regulated genes in volved in learning and memory are indicated in the plot. Genes are colour coded as stated in the legend. Both axes are log scaled.

**Figure 7 biomedicines-10-01946-f007:**
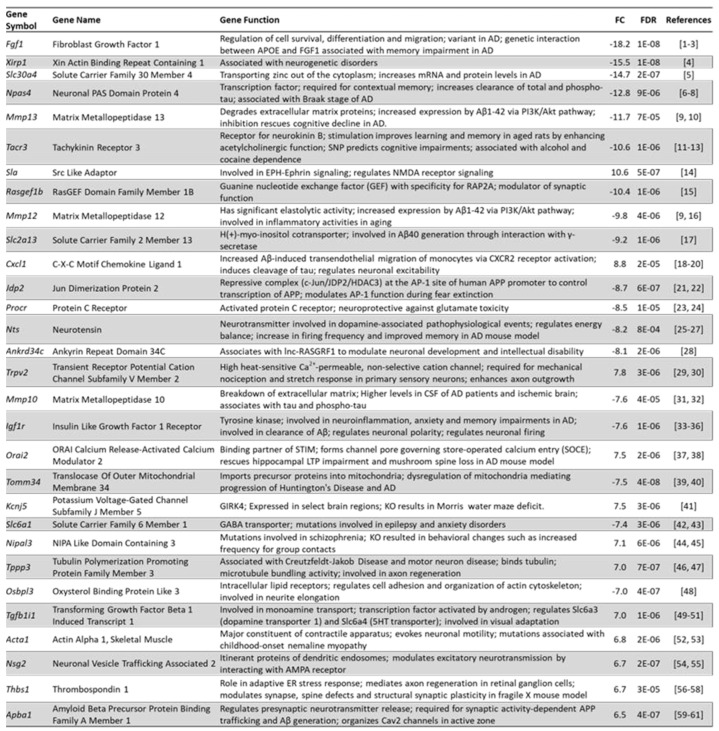
**Top ranking genes with neuronal functions**. APOE: apolipoprotein E; p-tau: phosphorylated tau; Aβ: Amyloid beta; PI3K/Akt: phosphatidylinositol 3-kinase/protein kinase B; CSCR2: C-X-C motif chemokine receptor 2; HDAC3: histone deacetylase 3; AP-1: activator protein 1; APP: amyloid precursor protein; CSF: cerebrospinal fluid; STIM: stromal interaction molecule; LTP: long term potentiation; GABA: γ-aminobutyric acid; KO: knockout; 5HT: 5-hydroxytryptamine; ER: endoplasmic reticulum; Cav2: neuronal voltage-gated calcium channels. **References**: 1–3 [82,83,84], 4 [85], 5 [86], 6–8 [87,88,89], 9,10 [90,91], 11–13 [92,93,94], 14 [95], 15 [96], 9,16 [90,97], 17 [98], 18–20 [99,100,101], 21,22 [102,103], 23,24 [104,105], 25–27 [106,107,108], 28 [109], 29,30 [110,111], 31,32 [112,113], 33–36 [114,115,116,117], 37,38 [118,119], 39,40 [120,121], 41 [122], 42,43 [123,124], 44,45 [125,126], 46,47 [127,128], 48 [129], 49–51 [130,131,132], 52,53 [133,134], 54,55 [135,136], 56–58 [137,138,139] 59–61 [140,141,142]. FC indicates fold change, shown as-1/FC when <1.

**Figure 8 biomedicines-10-01946-f008:**
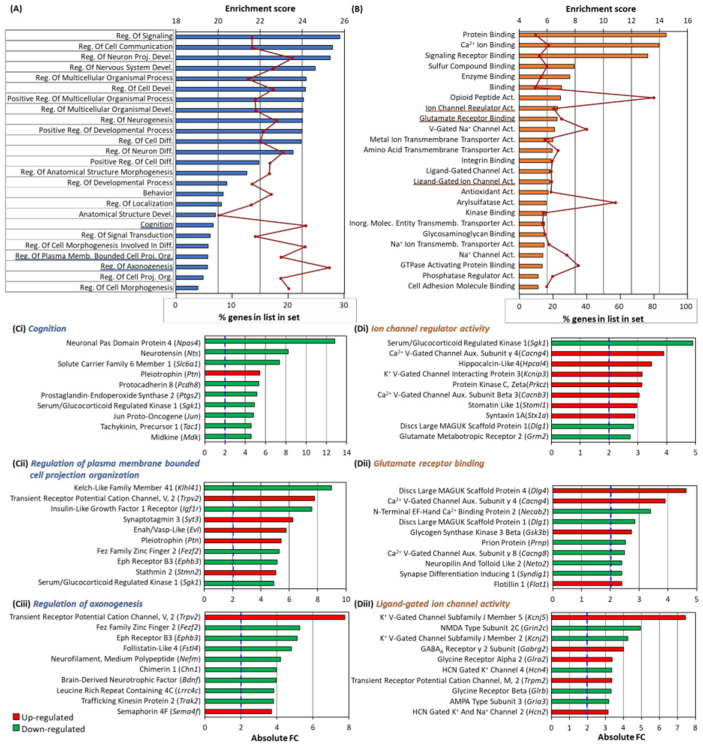
**GO analysis of Arc knock-down.** (**A**,**B**) Gene set enrichment analysis was performed to investigate the biological processes (**A**) and molecular functions (**B**) that the altered genes were involved in. The enrichment score is plotted against the category names. The enrichment score is the negative natural logarithm of the enrichment *p*-value derived from Fisher’s exact test and reflects the degree to which the gen e sets are overrepresented at the top or bottom of the entire ranked list of genes. Bars indicate the enrichment score while the line graph indicates the percentage of genes that are altered under the respective GO term. The top 25 biological processes (**A**) and molecular functions (**B**) are shown. Many of the categories are related to synaptic plasticity (underlined blue and orange) [20,21]. (**C**,**D**) Bar-charts showing genes involved in the stated category from Biological Processes (**Ci**–**Ciii**) and Molecular Functions (**Di**–**Diii**) and their respective fold changes. The top 10 regulated genes are shown. Dotted blue line indicates an absolute Fold Change of 2.

**Figure 9 biomedicines-10-01946-f009:**
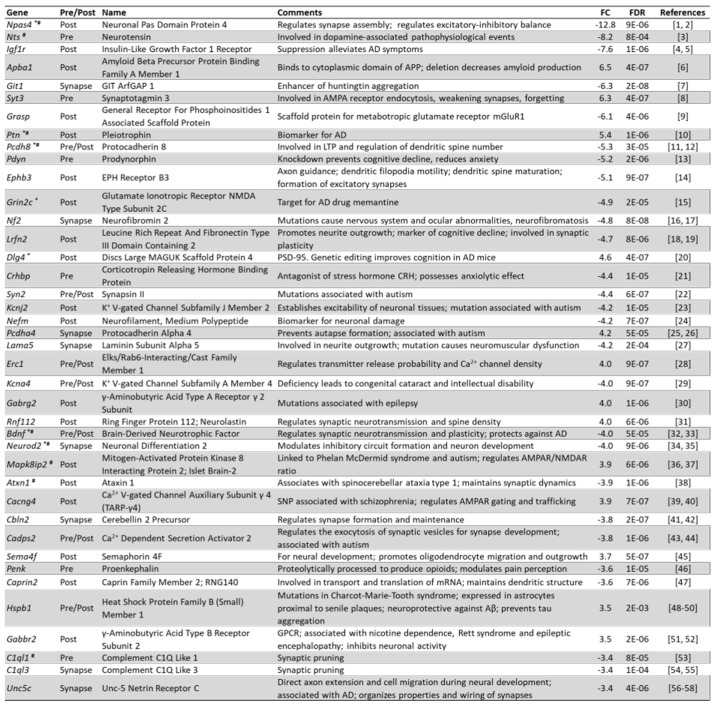
**Synaptic genes whose expression is regulated by Arc.** Shown are the top 40 synaptic genes (out of a total of 323) ranked by absolute fold change. Comments list relevant information about the function and disease association of the genes. An asterisk (*) indicates genes involved in neuroplasticity. A hashtag (#) indicates genes involved in cognition, learning and memory. APP: amyloid precursor protein; CRH: corticotropin-releasing hormone; V-gated: voltage-gated; TARP-γ4: transmembrane AMPR regulator protein γ4; Aβ: amyloid beta; GPCR: G-protein-coupled receptor. **References:** 1,2 [147,148], 3 [149]; 4,5 [114,115], 6 [142], 7 [150], 8 [151], 9 [152], 10 [153], 11,12 [154,155], 13 [156], 14 [157], 15 [158], 16,17 [159,160], 18,19 [161,162], 20 [163], 21 [164], 22 [165], 23 [166], 24 [167], 25,26 [168,169], 27 [170], 28 [171], 29 [172], 30 [173], 31 [174], 32,33 [175,176], 34,35 [177,178], 36,37 [179,180], 38 [181], 39,40 [182,183], 41,42 [184,185], 43,44 [186,187], 45 [188], 46 [189],47 [190], 48–50 [191,192,193], 51,52 [194,195], 53 [196], 54,55 [197,198], 56–58 [199,200,201].

**Figure 10 biomedicines-10-01946-f010:**
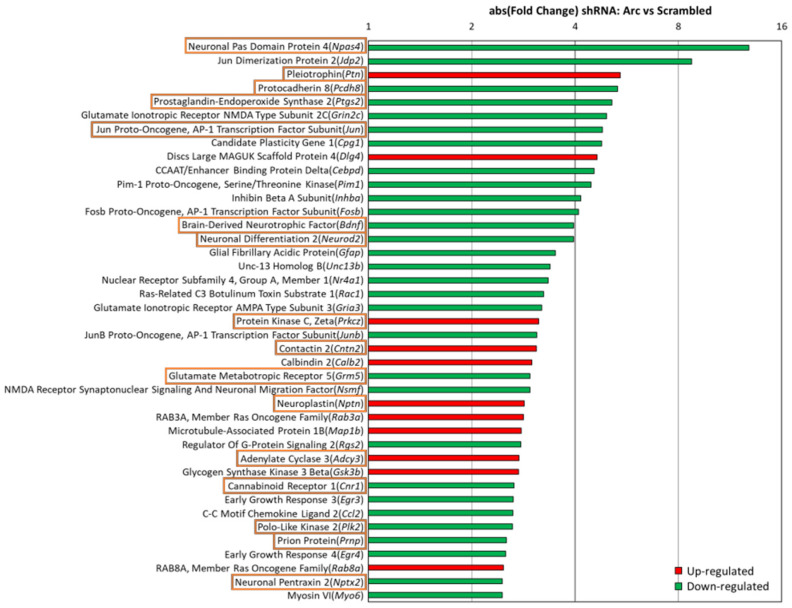
**Neuronal plasticity genes regulated by Arc.** Neuronal plasticity genes were manually curated in addition to reference to GO terminology from the gene ontology consortium. Neuronal plasticity genes with absolute FC ≥ 2.5 are shown. Genes that are involved in cognition or learning and memory are marked by orange boxes.

**Figure 11 biomedicines-10-01946-f011:**
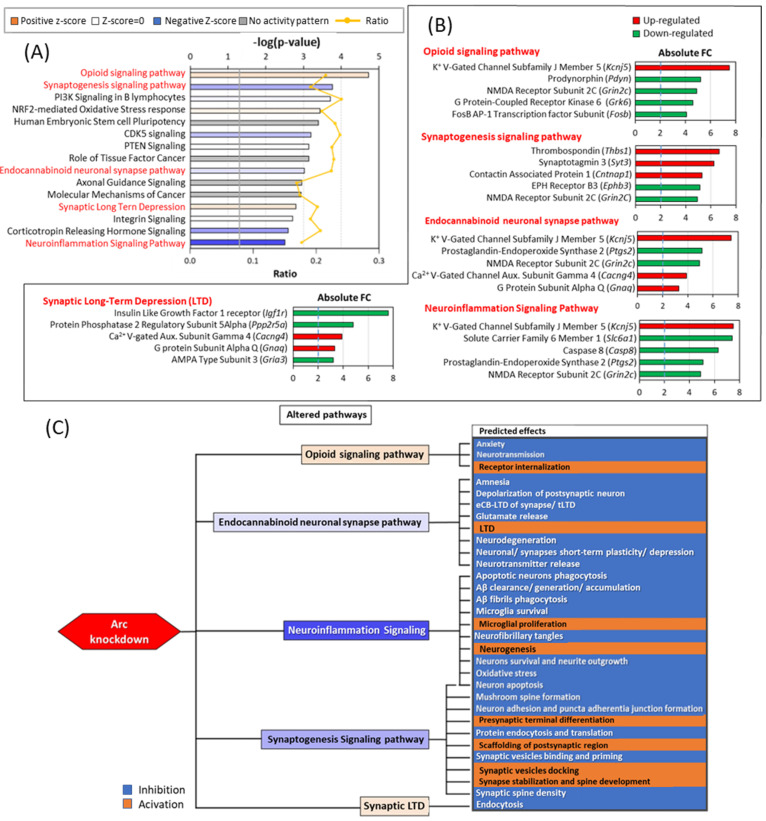
**Pathways altered by Arc knockdown.** (**A**) Bar chart showing altered pathways identified by IPA. The orange line graph indicates the ratio of genes that were involved in the specified pathway. The grey line indicates threshold at *p*-value 0.05. The orange and blue bars indicate predicted activation and inhibition of pathways, respectively (determined by z-score). (**B)** Top 15 significantly altered pathways are shown. Pathways with predicted activation or inhibition of downstream effects are in red, further elaborated in panel C. The top 5 genes altered in the respective pathways are shown on the right and bottom of the alter ed pathway bar chart. (**C**) Diagram describing the predicted effects of the altered pathways. Five pathways are highlighted, and the downstream effects as predicted by IPA are listed.

**Figure 12 biomedicines-10-01946-f012:**
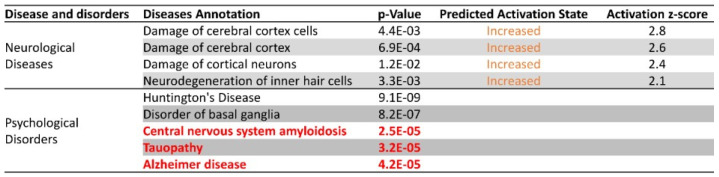
**Arc and neurological disorders**. Prevention of activity-dependent expression of Arc resulted in gene expression profile changes that are associated with neurological diseases and psychological disorders, including Huntington’s and Alzheimer’s disease, CNS amyloidosis and Tauopathy.

**Figure 13 biomedicines-10-01946-f013:**
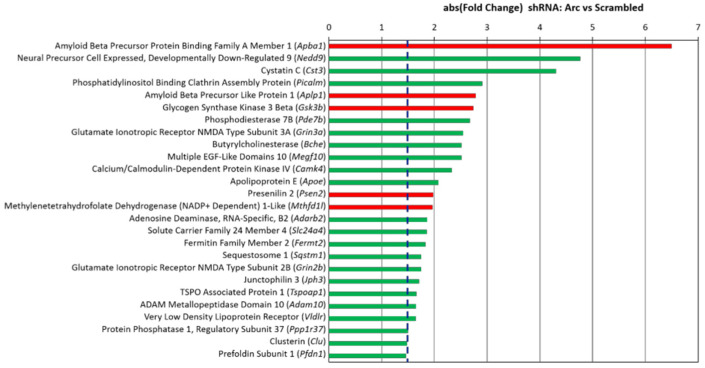
**Alzheimer’s susceptibility genes affected by Arc knockdown.** The expression levels of 26 AD susceptibility genes were affected when activity-dependent Arc expression was prevented by an shRNA. Green bars indicated that the mRNA level was downregulated, while red bars indicate upregulation. The blue line indicates an absolute fold change of 1.5.

**Figure 14 biomedicines-10-01946-f014:**
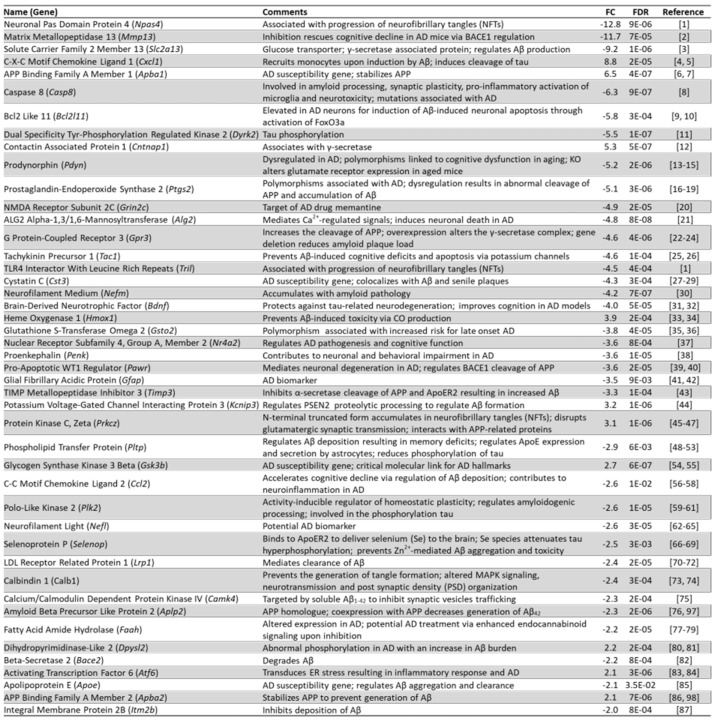
**Alzheimer’s genes regulated by Arc.** BACE1: β-secretase 1; APP: amyloid precursor protein; FoxO3a: forkhead box O3; ApoER2: apolipoprotein E receptor 2; PSEN2: presenilin 2; MAPK: mitogen-activated protein kinase; ER: endoplasmic reticulum. **References:** 1 [88], 2 [91], 3 [98], 4–5 [100,101], 6–7 [142,215], 8 [216], 9–10 [217,218], 11 [219], 12 [220], 13–15 [221,222,223], 16–19 [224,225,226,227], 20 [158], 21 [228], 22–24 [229,230,231], 25–26 [232,233], 27–29 [234,235,236], 30 [114], 31–32 [175,176], 33–34 [237,238], 35–36 [239,240], 37 [241], 38 [242], 39–40 [243,244], 41–42 [245,246], 43 [247], 44 [248], 45–47 [249,250,251], 48–53 [252,253,254,255,256,257], 54–55 [258,259], 56–58 [260,261,262], 59–61 [263,264,265], 62–65 [266,267,268,269], 66–69 [270,271,272,273], 70–72 [274,275,276], 73–74 [277,278], 75 [279], 76 [280], 77–79 [281,282,283], 80–81 [284,285], 82 [286], 83–84 [287,288], 85 [289], 86 [290], 87 [291].

**Figure 15 biomedicines-10-01946-f015:**
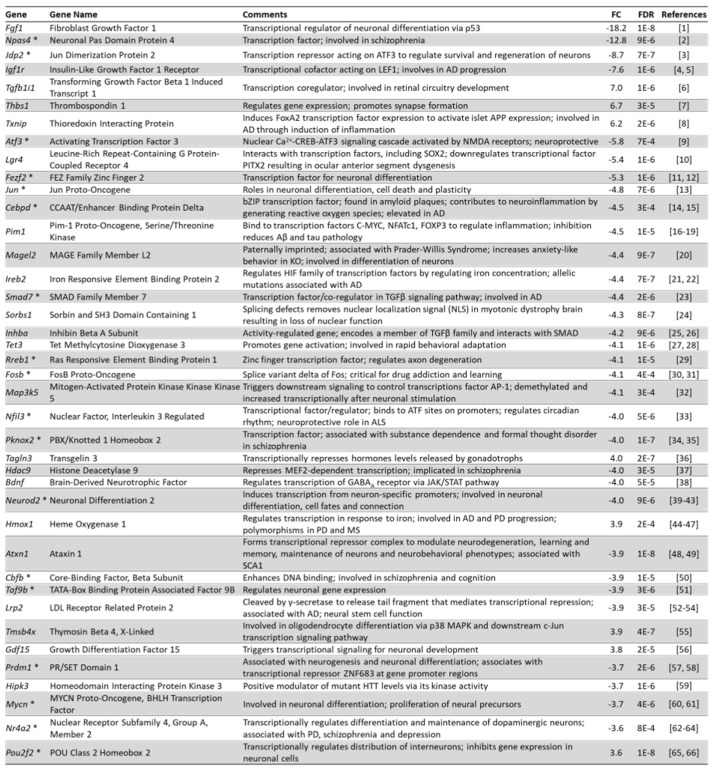
**Arc controls genes involved in transcriptional regulation.** Shown are the top 40 genes with neuronal relevance. An asterisk (*) indicates transcription factors. ATF3: activating transcription factor 3; LEF1: lymphoid enhancer binding factor 1; FoxA2: forkhead box A2; APP: amyloid precursor protein; CREB: cAMP response element-binding protein; SOX2: SRY-box 2; PITX2: paired like homeodomain 2; bZIP: basic leucine zipper domain; C-MYC: MYC proto-oncogene, BHLH transcription factor; NFATc1: nuclear factor of activated T cells 1; FOXP3: forkhead box P3; HIF: hypoxia inducible factor; TGFβ: transforming growth factor beta; PD: Parkinson’s disease; NLS: nuclear localisation signal; SMAD: transcription factors forming the core of the TGFβ signalling pathway; AP-1: activator protein 1; ALS: amyotrophic lateral sclerosis; MEF2: monocyte enhancer factor; GABA_A_: γ-aminobu- tyric acid type A; JAK/STAT: Janus kinases/ signal transducer and activator of transcription; SCA1: spinocerebellar ataxia type 1; MAPK: mitogen-activated protein kinase; ZNF683: zinc finger protein 683; HTT: huntingtin. **References:** 1 [292], 2 [293], 3 [294], 4–5 [114,295], 6 [132], 7 [296], 8 [297], 9 [298], 10 [299], 11–12 [300,301], 13 [302], 14–15 [303,304], 16–19 [305,306,307,308], 20 [309], 21–22 [310,311], 23 [312], 24 [313], 25–26 [314,315], 27–28 [316,317], 29 [318], 30–31 [319,320], 32 [321], 33 [322], 34–35 [323,324], 36 [325], 37 [326], 38 [327], 39–43 [177,178,328,329,330], 44–47 [331,332,333,334], 48–49 [335,336], 50 [337], 51 [338], 52–54 [339,340,341], 55 [342], 56 [343], 57–58 [344,345], 59 [346], 60–61 [347,348], 62–64 [349,350,351], 65, 66 [352,353].

**Figure 16 biomedicines-10-01946-f016:**
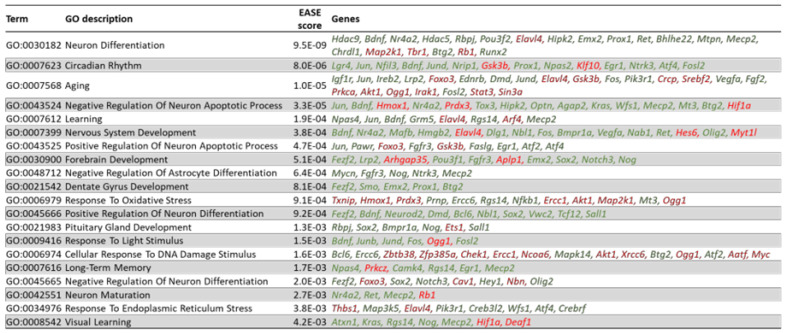
**Biological processes controlled by the Arc-dependent transcriptional regulators.** The top 20 biological processes of neurological relevance are listed. EASE score is a modified Fisher exact p-value measuring the gene-enrichment in the annotated terms. Genes are arranged in order of highest to lowest absolute fold change. Genes highlighted in red are upregulated, while those highlighted in green are downregulated.

**Figure 17 biomedicines-10-01946-f017:**
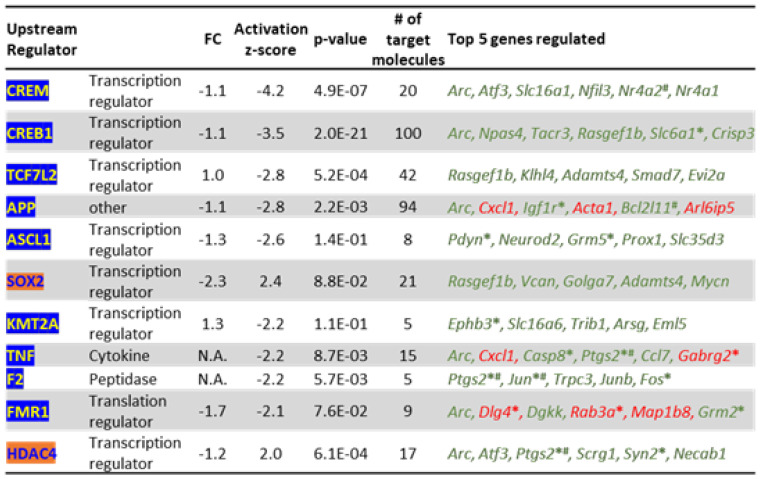
**Upstream regulators associated with differential gene expression observed upon knockdown of Arc.** Activation z-score indicates the predicted activity of upstream regulators by IPA analysis. Upstream regulators that were predicted to be inhibited are highlighted in blue while those activated are highlighted in orange. Regulated genes were highlighted in green (downregulated) and red (upregulated).

**Figure 18 biomedicines-10-01946-f018:**
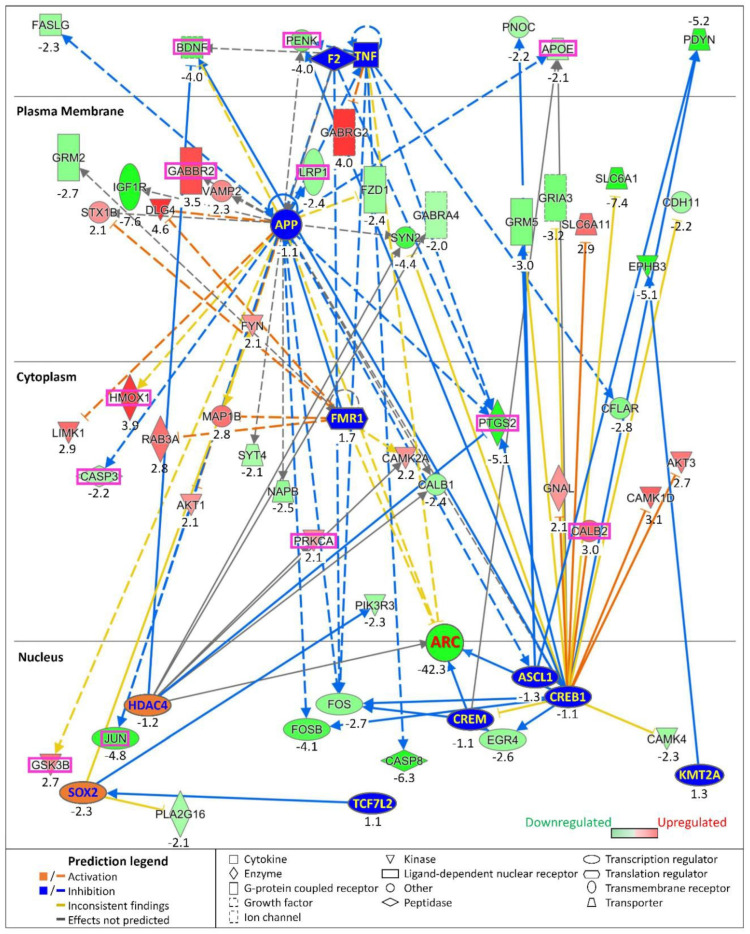
**Upstream regulators of differential gene expression caused by Arc knockdown.** The map shows 11 upstream regulators (blue and orange boxes) predicted by IPA to mediate altered gene expression upon Arc knockdown. Genes are positioned in the extracellular space, the plasma membrane, the cytosol, or the nucleus, depending on where their associated proteins are located. Arc was positioned at the interface of nucleus and cytoplasm because it can be in either compartment. Only genes that were involved in the following pathways and disease annotations are shown: (i) opioid signalling, (ii) synaptogenesis, (iii) endocannabinoid neuronal synapse pathway, (iv) synaptic LTD, (v) neuroinflammation, (vi) CNS amyloidosis, (vii) tauopathy and (viii) AD. Genes associated with disease annotations are boxed in magenta. The respective fold changes are indicated below each gene.

**Figure 19 biomedicines-10-01946-f019:**
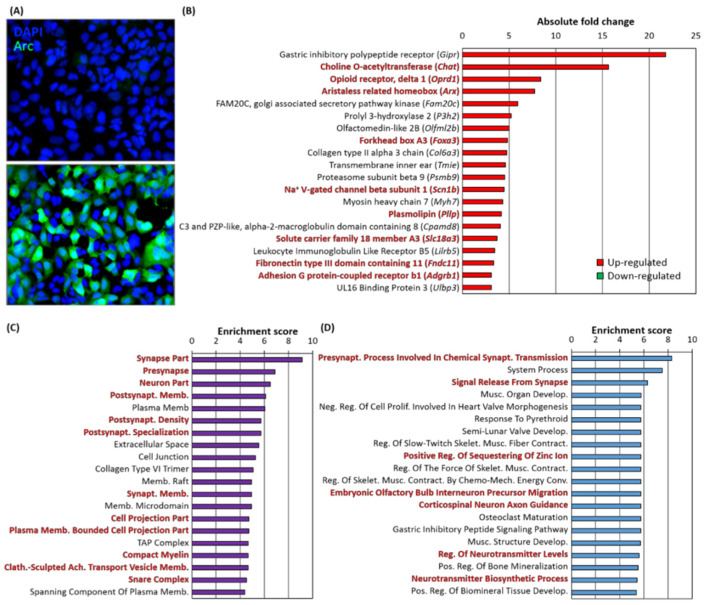
**HEK293T cells exhibit neuronal properties upon induced expression of Arc.** (**A**) Endogenous Arc expression was enhanced in HEK293T cells by targeting two single guide RNAs (sgRNAs) containing MS2 aptamers to the Arc promoter with the CRISPR/Cas9 Synergistic Activation Mediator system (see Methods for details). As a negative control, we used two sgRNAs targeting the promoter of the lac operon. Control cells (**top**) and Arc-induced cells (**bottom**) were stained for Arc (green) and DNA was labelled with DAPI (blue). About 90% of the cells expressed Arc. Scale bar is 30 µm. (**B**) Graph showing the top 20 differentially expressed genes upon the induction of endogenous Arc in HEK293T cells. RNA-Seq was used to compare the mRNA levels between the Arc-induced and control HEK293T cells. Neuronal genes are bolded and highlighted in red. (**C**,**D**) GO analysis of the differential expressed genes upon overexpression of Arc. The top 20 cellular components (**C**) and biological processes (**D**) are presented. Neuronal features were bolded and highlighted in red. Na^+^: sodium; V-gated: voltage-gated; Postsynapt: postsynaptic; Memb: membrane; Synapt: synaptic; Clath: clathrin; Ach: acetylcholine; Presynapt: presynaptic; Musc: muscle; Develop: development; Neg: negative; Reg: regulation; Prolif: proliferation; Skelet: skeletal; Contract: contraction and Mech: mechanical; Conv: conversion.

**Figure 20 biomedicines-10-01946-f020:**
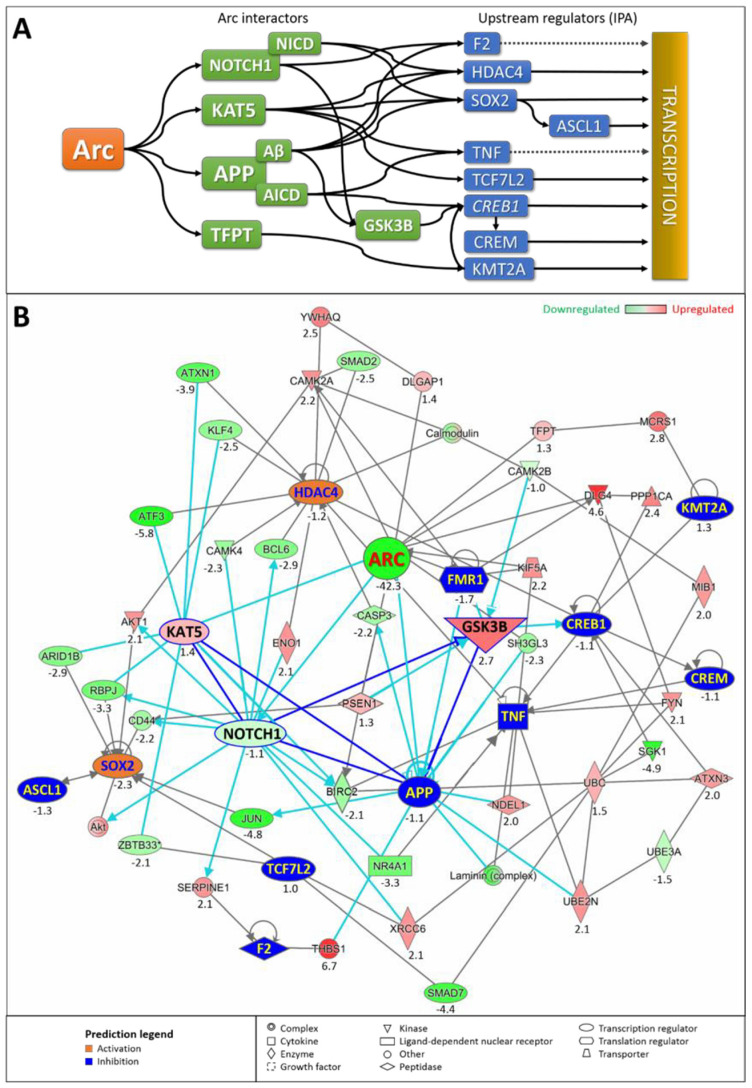
**Regulation of upstream regulators by Arc interactors.** (**A**) Schematic diagram illustrating how interactors of Arc could bring about changes in activity of upstream regulators identified by IPA, which in turn results in alteration of gene transcription. The dotted line indicates an indirect effect on transcription through the regulation of a transduction cascade. (**B**) Diagram showing the functional connectivity between Arc interactors (KAT5, NOTCH1, GSK3B and APP), upstream regulators highlighted in orange (activated) or blue (inhibited) and genes that mediate their interaction. Connections of Arc interactors with other genes are highlighted in cyan. Connections between KAT5, NOTCH1, GSK3B and APP are highlighted in blue. Genes from A are shown in bold.

## Data Availability

Not applicable.

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
