# Peer review of "Arc Regulates Transcription of Genes for Plasticity, Excitability and Alzheimer’s Disease"

_biomedicines, 2022, doi:10.3390/biomedicines10081946_

Round 1

Reviewer 1 Report

This is an ambitious report summarizing and adding to knowledge of Arc in relation to neuron function, with a focus on Alzheimer’s disease. 

Section 1, paragraph 1: Suggest you break up this long paragraph for easier reading.

Section 1: Change "in upregulated" to "is upregulated".

Section 1: Change "hairpin RNA," to "hairpin RNA".

Section 2.2: Change "Hippocampal" to "Hippocampi".

Section 2.4: Change "1% (v/v)," to "1% (v/v)".

Section 2.4: You mention dyes presumably coupled to secondary antibodies, but what secondary antibodies did you use?

Section 2.4: For NaBH4, use subscript font for 4.

Section 2.5: Why did you use a different buffer to anneal Arc shRNAs than to anneal scrambled shRNAs?

Section 2.5: Change "DIV19-22" to "DIV19 and DIV22".

Section 2.7: Why use averages of Arc intensity from non-4BF controls as the cut-off threshold? I'd think it would be the baseline rather than the cut-off. Shouldn't the cut-off be between the Arc intensity of non-4BF and 4BF cells?

Section 2.9: Change "are" to "were".

Section 2.12: Why use an absolute fold change of 2.00 as the cut-off? It seems arbitrary. Why not 1.50 or 3.25, or whatever value 0.05 of genes exceed? Why use a cut-off at all? Would it be more informative to weight genes by the degree of fold change rather than to lump them into either one of two bins (altered expression or unaltered expression)? Maybe this method of dichotomizing is just the standard, in which case you can say so.

Section 2.13: Use the Greek letter micron instead of "u" in "ug/ml".

Figure 2A: Change "Heat maps of the relative DAPI intensity of the nucleus is" to "Heat maps of the relative DAPI intensity of the nucleus are".

Figure 2B: Change "relative homogenous" to "relatively homogenous".

Figure 2B: Change "neurons (left panel) was" to "neurons (left panel) were".

Figure 2B: Change "indication" to "indicating".

Section 3.2: I click on "Movie 1" but cannot see a video. CTRL Click also does not work. I tried 3 different pdf viewers.

Figure 5: Show "A" and "B" labels and the scale bar.

Figure 5: Section 2.9 suggests that Arc would appear green and H3K9Ac-S10P red. The last sentence in the Figure 5 caption is consistent with this. But the labels in Figure 5 show the reverse. Please check. 

Figure 5: Will regions of overlap appear yellow? If so, comment on this in the figure.

Section 3.5, Sentence 2: "4BF" should be bold.

Section 3.5: Change "is" to "was".

Figure 6: Please check the x-axis numbers. I think your numbers correctly indicate the fold change in the positive direction but not in the negative direction. In the negative direction, I suppose you mean 1/64, 1/32, etc., rather than -64, -32, etc. If you want to use negative and positive numbers, then change the axis label to log(base 2) fold change, in which case I assume that your labels would become -6, -5, -4, -3, -2, -1, 0, 1, 2, 3, 4, 5, and 6.

Figure 6: Some genes are underlined in red (I think. I'm a bit colorblind, so it's hard for me to distinguish some of these colors.). Indicate in the legend what red underline means.

Section 3.5: Blocking Arc expression changed levels of ~1900 genes. That's a large fraction of the total number of human genes. It's about a tenth of all genes. So if you find that levels of some members of a class of genes are altered, that could be a coincidence because levels of so many genes are altered. To make a case that it's meaningful or significant that levels of a class of genes (such as ion channels) are altered, you should use statistical tests. I suppose chi-square may be appropriate. For example, you'd count the number of ion channel genes that are altered, the number of ion channel genes that are NOT altered, the number of all genes (other than ion channels) that are altered (~1900), and the number of all genes (other than ion channels) that are not altered (~18,000), and then plug them into a chi-square test and show us the p value. Since you're looking at multiple families of genes, you should also correct for multiple comparisons, using Bonferroni or some other method. If it turns out that the gene families you examine all fail these tests, I suppose that you could still make a case that even if expression levels of a gene family were not altered any more than that of a random set of genes, Arc nevertheless affects that gene family and its functions, but you should discuss this.

Section 3.5: According to Section 2.11, you sequenced 3 sets of cells (9 samples). But in Section 3.5, you only reported 2 sets. How did you analyze cells treated for 8 h with 4BF but without shRNA?

Table 1: Change reference numbers in the table itself to match those in the reference list at the end of the paper.

Section 3.5, paragraph 2: Check the FC values in Table 1 (see one of the comments above, on Figure 6 FC values). I think they may not be FC, but may be 2xlog(base 2) changes.

Figure 7 caption: Change "The top 25 biological processes (B) and molecular functions (C) are shown.” to “The top 25 biological processes (A) and molecular functions (B) are shown.”

Figure 7: Why are certain categories underlined, but not others? One could argue that other categories are also related to synaptic plasticity. It seems like you selected these for highlighting somewhat arbitrarily. That’s OK, but you should explain.

Figure 7: Check the FC values, as in Figure 6.

Section 3.7: The beginning of this section repeats parts of Section 3.6. Re-write to integrate these two sections better.

Section 3.7: Re-write the sentence that starts with “These genes”. It is confusing.

Table 2: As with previous comments, check the FC values, and convert the reference numbers.

Figure 9: To make it easier to talk about the graphs in (A), re-name the first graph (A), the other 5 graphs (B), and the last graph (C). 

Figure 9: I don’t understand the colors in the first graph of (A). It looks like there are about 7 colors for the bars but only 4 colors in the legend. Please simplify. OK, after spending a long time reading and staring at the first graph of (A), I think I finally understand. You meant to have the redness of the bars indicate degree of activation, and blueness indicate degree of inhibition, and gray (which also looks red to me because of my partial color-blindness) indicate no activity pattern. Maybe non-colorblind people can see this readily, but it seems pretty confusing. I suggest a simpler way to portray this information is to show z-score by length of the bar to the left or right of a vertical line, and the -log(p-value) by gray intensity of the bar. However, some pathways have no z-score, so instead of gray intensity, you may have to just list a number in a column next to the pathway name. For no activity pattern, show no bar and say “No activity pattern” or something like that where the bar would have been. Alternatively, you may keep the current bar graph but replace the colors with a column of numbers indicating the z-scores. Another alternative is to keep everything as it is, but make the colors easier to distinguish by changing the color for “No activity pattern” to black or vertical-striped, and explain better that the intensity of the red or blue colors is proportional to the z-score. Ah, now that I see the final graph (B), I realize that the red or blue colors are needed here. Thus you may use the last alternative for the first graph in (A).

Figure 9A: Why did you pick the 5 pathways that you highlighted in red? Was there a quantitative criterion? Or was it qualitative? Please elaborate in the text and in the caption.

Figure 9 caption: Change “its” to “the”.

Figure 9B: You show that both Neuroinflammation and Synaptogenesis lead to Neuron apoptosis. Is that correct?

Figure 9B: Describe in either the text or the caption the criteria to select the predicted effects. Are these all the predicted effects that passed a p-value threshold? Are they only some of the predicted effects, based on qualitative criteria (they seemed interesting)?

Table 3: Indicate in the caption what red bold text means.

Section 3.9: 26 of the 39 AD genes seems very significant. Please show the statistics.

Section 3.10: Delete “as such”.

Section 3.10: Change “(Table 6,” to “(Table 5),”.

Figure 11: I don’t see the upstream regulators in blue and orange boxes. They are in different shapes. It is hard to see them. The figure is quite busy, making it difficult to read and interpret. But I’m not sure how to suggest to improve its readability besides splitting it into two figures, each conveying part of the information.

Figure 12B: You picked the top 20 apparently based on the FC value, thus they must be up-regulated. There’s therefore no point to include down-regulated (green) in the legend, or any legend at all.

Figure 13: The orange or blue highlighting is not obvious.

Figure 13: Delete “highlighted are ”.

Figure 13: Why do only some connections have arrows?

Figure 13: You should state in the legend that genes from (A) are shown in bold.

Author Response

We would like to thank the reviewer for the very thorough report and constructive critique.

Section 1, paragraph 1: Suggest you break up this long paragraph for easier reading. Done

Section 1: Change "in upregulated" to "is upregulated". Done

Section 1: Change "hairpin RNA," to "hairpin RNA". Done

Section 2.2: Change "Hippocampal" to "Hippocampi". Done

Section 2.4: Change "1% (v/v)," to "1% (v/v)". Done

Section 2.4: You mention dyes presumably coupled to secondary antibodies, but what secondary antibodies did you use? Anti-mouse. Section 2.4 was updated.

Section 2.4: For NaBH4, use subscript font for 4. Done

Section 2.5: Why did you use a different buffer to anneal Arc shRNAs than to anneal scrambled shRNAs? Both buffers were used for annealing the Arc shRNAs and scrambled shRNAs. Upon analysis, Buffer A works for Arc shRNA and buffer B works for scrambled shRNA. Edits are included in manuscript.

Section 2.5: Change "DIV19-22" to "DIV19 and DIV22". Done

Section 2.7: Why use averages of Arc intensity from non-4BF controls as the cut-off threshold? I'd think it would be the baseline rather than the cut-off. Shouldn't the cut-off be between the Arc intensity of non-4BF and 4BF cells?  In cell cultures treated with 4BF, approximately half the neurons will be induced to express Arc. In this population, Arc intensity varies greatly, with some neurons reaching very high expression levels (population A). while others show lower expression (population B), albeit still higher than the intensity seen for the non-4BF control. Such gradual differences in population B, compared to non-4BF, is difficult to determine at times; it is very dependent on the culture or other extraneous variables. As such, a clear cut-off will be difficult to determine.

Therefore, to circumvent such observations and provide the best comparison between non-4BF treated and 4BF-treated cells, the average of Arc intensity was taken from the non-4BF, since Arc expression was only observed upon stimulation and acts in many instances as molecular markers for activated neurons 1,2.  Any intensity higher than (non-4BF) control could then be taken as an induction of Arc.

Explanations are now included in the revised manuscript.

Section 2.9: Change "are" to "were". Done

Section 2.12: Why use an absolute fold change of 2.00 as the cut-off? It seems arbitrary. Why not 1.50 or 3.25, or whatever value 0.05 of genes exceed? Why use a cut-off at all? Would it be more informative to weight genes by the degree of fold change rather than to lump them into either one of two bins (altered expression or unaltered expression)? Maybe this method of dichotomizing is just the standard, in which case you can say so.  Differential gene expression was considered first, on the basis that they are statistically significant (FDR step-up < 0.05). A second criterium was added that the absolute fold change is at least 2. This is to focus on genes whose expression change is not only statistically significant, but also biologically relevant. A 100% increase/decrease in transcription level, if statistically significant, should be biologically relevant. Adding a fold change criterium limits the number of differentially expressed genes for further analysis. For example, in our dataset: for absolute fold change 1.5, there are 4537 differentially expressed genes compared to absolute fold change of 2 with 1945 genes3. With the FC>2 dataset, one could focus on those genes with a higher magnitude of change. To further improve biological relevance, the list of differentially expressed genes with FC>2 was extracted and put through a gene ontology enrichment analysis to provide a more coherent insight on the biological relevance and identify gene ontology terms that are over-represented upon a change in expression of Arc3-5.

Explanations are now elaborated in the updated manuscript.

Section 2.13: Use the Greek letter micron instead of "u" in "ug/ml". Done

Figure 2A: Change "Heat maps of the relative DAPI intensity of the nucleus is" to "Heat maps of the relative DAPI intensity of the nucleus are". Done

Figure 2B: Change "relative homogenous" to "relatively homogenous". Done

Figure 2B: Change "neurons (left panel) was" to "neurons (left panel) were". Done

Figure 2B: Change "indication" to "indicating". Done

Section 3.2: I click on "Movie 1" but cannot see a video. CTRL Click also does not work. I tried 3 different pdf viewers. The movie will be hosted by the publisher, instead of YouTube, which will fix this problem.

Figure 5: Show "A" and "B" labels and the scale bar. Done

Figure 5: Section 2.9 suggests that Arc would appear green and H3K9Ac-S10P red. The last sentence in the Figure 5 caption is consistent with this. But the labels in Figure 5 show the reverse. Please check. Arc is shown in red and H3K8Ac-S10P is shown in green, as stated. The arrows are placed inside the Arc puncta. We have changed the wording of the last sentence to make this clear.

Figure 5: Will regions of overlap appear yellow? If so, comment on this in the figure. Done

Section 3.5, Sentence 2: "4BF" should be bold. Done

Section 3.5: Change "is" to "was". Done

Figure 6: Please check the x-axis numbers. I think your numbers correctly indicate the fold change in the positive direction but not in the negative direction. In the negative direction, I suppose you mean 1/64, 1/32, etc., rather than -64, -32, etc. If you want to use negative and positive numbers, then change the axis label to log(base 2) fold change, in which case I assume that your labels would become -6, -5, -4, -3, -2, -1, 0, 1, 2, 3, 4, 5, and 6. We apologize for not having this explained properly. We have converted the Fold Change as follows: if smaller then 1, use -1/FC, if 1 or larger use FC. We have explained this now in the Methods section.

Figure 6: Some genes are underlined in red (I think. I'm a bit colorblind, so it's hard for me to distinguish some of these colors.). Indicate in the legend what red underline means. No genes are underlined in red. Some are underlined in green, indicating they are cognitive genes. This is already shown in the legend.

Section 3.5: Blocking Arc expression changed levels of ~1900 genes. That's a large fraction of the total number of human genes. It's about a tenth of all genes. So if you find that levels of some members of a class of genes are altered, that could be a coincidence because levels of so many genes are altered. To make a case that it's meaningful or significant that levels of a class of genes (such as ion channels) are altered, you should use statistical tests. I suppose chi-square may be appropriate. For example, you'd count the number of ion channel genes that are altered, the number of ion channel genes that are NOT altered, the number of all genes (other than ion channels) that are altered (~1900), and the number of all genes (other than ion channels) that are not altered (~18,000), and then plug them into a chi-square test and show us the p value. Since you're looking at multiple families of genes, you should also correct for multiple comparisons, using Bonferroni or some other method. If it turns out that the gene families you examine all fail these tests, I suppose that you could still make a case that even if expression levels of a gene family were not altered any more than that of a random set of genes, Arc nevertheless affects that gene family and its functions, but you should discuss this. We were also worried about that. For some classes of genes, you can get accurate numbers for how many there are in the genome. Below is a table that lists the number of genes for a class in the human genome, the number of genes altered by Arc, and what percentage was altered by Arc. For each gene class the number is significantly larger than 10%. So, these genes did not end up in the list of affected genes solely by chance.

Class

Genome

Arc

Percentage

Na, K , Ca channels

145

33

23%

Neurotransmitter receptors

58

14

24%

Trp channels

28

7

25%

Transporters

325

139

43%

Section 3.5: According to Section 2.11, you sequenced 3 sets of cells (9 samples). But in Section 3.5, you only reported 2 sets. How did you analyze cells treated for 8 h with 4BF but without shRNA?  The cells for 8 h with 4BF were analyzed with 0 h 4BF (without shRNA) in an RT-PCR setting to ensure there is Arc induction. The same batch of cells with 8 h 4BF, 8 h Arc shRNA, and 8 h Arc scrambled shRNA were then sent for RNA-sequencing. 3 sets of such experiments were sent for RNA-sequencing as indicated in section 2.11. Only the results for Arc shRNA and Arc scrambled shRNA were shown in section 3.5 as this was of the utmost concern to show the differential expression in instances with Arc expression (Arc scrambled shRNA) and without Arc expression (Arc shRNA). A gene differential analysis was also performed for i) 8 h 4BF with 8 h 4BF Arc scrambled shRNA and ii) 8 h 4BF with 8 h 4BF Arc shRNA using the same conditions indicated (data not presented).

Elaboration is included in the updated manuscript.

Table 1: Change reference numbers in the table itself to match those in the reference list at the end of the paper. That will be very difficult and invite mistakes. References are managed by EndNote 11 and the Table resides in Excel.  Endnote cannot create or update references in Excel. We realize this is cumbersome, but it can’t be helped.

Section 3.5, paragraph 2: Check the FC values in Table 1 (see one of the comments above, on Figure 6 FC values). I think they may not be FC, but may be 2xlog(base 2) changes. We have amended the legend to make this clear.

Figure 7 caption: Change "The top 25 biological processes (B) and molecular functions (C) are shown.” to “The top 25 biological processes (A) and molecular functions (B) are shown.” Done

Figure 7: Why are certain categories underlined, but not others? One could argue that other categories are also related to synaptic plasticity. It seems like you selected these for highlighting somewhat arbitrarily. That’s OK, but you should explain.  Yes, as indicated in the text, the top 25 biological processes and molecular functions were shown. In these top 25, many others are also related to synaptic plasticity. I have underlined 3 examples from the biological processes (in blue) and 3 examples of molecular function (in orange). These examples are known examples of plasticity. I have amended and included the references in the updated text.

Figure 7: Check the FC values, as in Figure 6. Done

Section 3.7: The beginning of this section repeats parts of Section 3.6. Re-write to integrate these two sections better. We have rewritten those sections.

Section 3.7: Re-write the sentence that starts with “These genes”. It is confusing. We have rewritten that sentence

Table 2: As with previous comments, check the FC values, and convert the reference numbers. Done

Figure 9: To make it easier to talk about the graphs in (A), re-name the first graph (A), the other 5 graphs (B), and the last graph (C). Done

Figure 9: I don’t understand the colors in the first graph of (A). It looks like there are about 7 colors for the bars but only 4 colors in the legend. Please simplify. OK, after spending a long time reading and staring at the first graph of (A), I think I finally understand. You meant to have the redness of the bars indicate degree of activation, and blueness indicate degree of inhibition, and gray (which also looks red to me because of my partial color-blindness) indicate no activity pattern. Maybe non-colorblind people can see this readily, but it seems pretty confusing. I suggest a simpler way to portray this information is to show z-score by length of the bar to the left or right of a vertical line, and the -log(p-value) by gray intensity of the bar. However, some pathways have no z-score, so instead of gray intensity, you may have to just list a number in a column next to the pathway name. For no activity pattern, show no bar and say “No activity pattern” or something like that where the bar would have been. Alternatively, you may keep the current bar graph but replace the colors with a column of numbers indicating the z-scores.

Another alternative is to keep everything as it is, but make the colors easier to distinguish by changing the color for “No activity pattern” to black or vertical-striped, and explain better that the intensity of the red or blue colors is proportional to the z-score.   Done

Ah, now that I see the final graph (B), I realize that the red or blue colors are needed here. Thus you may use the last alternative for the first graph in (A).

Figure 9A: Why did you pick the 5 pathways that you highlighted in red? Was there a quantitative criterion? Or was it qualitative? Please elaborate in the text and in the caption. The 5 pathways are provided by the IPA analysis which, based on the differential gene expression list, predicted that these 5 pathways could result in a downstream activation or inhibition.

This is now elaborated in the text.

Figure 9 caption: Change “its” to “the”. Done

Figure 9B: You show that both Neuroinflammation and Synaptogenesis lead to Neuron apoptosis. Is that correct? That is correct. Based on the IPA analysis, under synaptogenesis signaling pathway and neuroinflammation signaling pathway, neuron apoptosis is one of their predicted effects.

Figure 9B: Describe in either the text or the caption the criteria to select the predicted effects. Are these all the predicted effects that passed a p-value threshold? Are they only some of the predicted effects, based on qualitative criteria (they seemed interesting)? These pathways are selected as they are indicated in the IPA analysis output that the pathways could result in an activation or inhibition of downstream effects. The downstream effects are illustrated in Figure 9B. Better elaborations are included in the text.

Table 3: Indicate in the caption what red bold text means. Done

Section 3.9: 26 of the 39 AD genes seems very significant. Please show the statistics. I have included the values of the FDR step-up for each of the gene in the table. The caption has been edited and amended.

Section 3.10: Delete “as such”. Done.

Section 3.10: Change “(Table 6,” to “(Table 5),”. Done

Figure 11: I don’t see the upstream regulators in blue and orange boxes. They are in different shapes. It is hard to see them. The figure is quite busy, making it difficult to read and interpret. But I’m not sure how to suggest to improve its readability besides splitting it into two figures, each conveying part of the information. It could be difficult to split into two figures as most of the mediators and regulators are interconnected. However, I have highlighted in the caption which are the blue and orange upstream regulators.

Figure 12B: You picked the top 20 apparently based on the FC value, thus they must be up-regulated. There’s therefore no point to include down-regulated (green) in the legend, or any legend at all. We have removed the reference to “down-regulated (green)”  from the legend.

Figure 13: The orange or blue highlighting is not obvious. We have changed the color of orange to make it more obvious. In addition, I have included in the caption, which are the activated regulators (in orange) and which are the inhibited regulators (in blue).

Figure 13: Delete “highlighted are ”. Done

Figure 13: Why do only some connections have arrows?  While arrowheads indicate an activating/inhibiting  relationship (sharp/blunt arrow), connectors without arrows represent a relationship between the two biomolecules, based on literature citations, of i) an interacting nature, or that ii) they could be upstream or downstream from each other. We have explained this in the caption.

Figure 13: You should state in the legend that genes from (A) are shown in bold. Done.

Submission Date

22 June 2022

Date of this review

29 Jun 2022 18:20:11

Bottom of Form

1            Gouty-Colomer, L. A. et al. Arc expression identifies the lateral amygdala fear memory trace. Mol Psychiatry 21, 1153, doi:10.1038/mp.2016.91 (2016).

2            Minatohara, K., Akiyoshi, M. & Okuno, H. Role of Immediate-Early Genes in Synaptic Plasticity and Neuronal Ensembles Underlying the Memory Trace. Front Mol Neurosci 8, 78, doi:10.3389/fnmol.2015.00078 (2015).

3            Koch, C. M. et al. A Beginner's Guide to Analysis of RNA Sequencing Data. Am J Respir Cell Mol Biol 59, 145-157, doi:10.1165/rcmb.2017-0430TR (2018).

4            Ashburner, M. et al. Gene ontology: tool for the unification of biology. The Gene Ontology Consortium. Nat Genet 25, 25-29, doi:10.1038/75556 (2000).

5            The Gene Ontology, C. The Gene Ontology Resource: 20 years and still GOing strong. Nucleic Acids Res 47, D330-D338, doi:10.1093/nar/gky1055 (2019).

Reviewer 2 Report

This is a nice work including the study of immediate-early gene Arc and its association with AD.

Being a severe disease, any solid contribution to better understanding the triggers, progression and therefore right decisions for treatment of AD should be valued.

The Arc relation and interaction to chromatin structure were studied in different neuron cells.

Different approaches such as fluorescent microscopy and RNA-seq were used.

For bioinformatic analysis, GO and IPA tools were utilised.

Ultimately, I can conclude that the work is exhaustive including the experimental part. The FM figures are nicely displayed as well. English is acceptable and the topic is highly actual.

Author Response

Thank you.

Reviewer 3 Report

The authors investigated how Arc regulates transcription of genes involved in neuronal plasticity, excitability, and Alzheimer’s Disease. The observed an increase in Arc expression following the treatment of hippocampal or cortical neuronal cultures with combined 4AP, bicuculline and forskolin. They reported that increase in Arc expression modulates several genes including those involved in the pathophysiology of AD. The results are relevant and important with a potential for high impact. I have the following concerns:

1.      How did the authors arrive at the concentrations of 4AP, bicuculline and forskolin used in this investigation?

2.      How did the authors know that the combinations of these drugs have synergistic effects on Arc expression?

3.      Did the authors try using agonist of these drugs to see whether Arc expression will be abolished?

Author Response

The authors investigated how Arc regulates transcription of genes involved in neuronal plasticity, excitability, and Alzheimer’s Disease. The observed an increase in Arc expression following the treatment of hippocampal or cortical neuronal cultures with combined 4AP, bicuculline and forskolin. They reported that increase in Arc expression modulates several genes including those involved in the pathophysiology of AD. The results are relevant and important with a potential for high impact. I have the following concerns:

  1. How did the authors arrive at the concentrations of 4AP, bicuculline and forskolin used in this investigation?

We have optimized the concentrations for these 3 drugs in early experiments and published them in previous papers [1-4]. We have updated the Methods section to add these citations.

  1. How did the authors know that the combinations of these drugs have synergistic effects on Arc expression?

The combination of 4AP and bicuculline has first been used to increase synaptic strength in cultured neurons by Hilmar Bading’s laboratory [5,6]. We have shown previously that forskolin rescues the mRNA translation impediment seen for Arc in cultured neurons [1]. The combination of these three drugs results in maximal induction of Arc expression.

  1. Did the authors try using agonist of these drugs to see whether Arc expression will be abolished?

That is an interesting idea. We have not tested that yet. 4AP blocks A-type K channels. I don’t know of any agonist for those family of K channels.

Bicuculline blocks GABA_A receptors. There are positive allosteric modulators for GABA_A receptors, and several are approved by the FDA  (e.g., Valium, Ambien), so those could be tested to see whether they inhibit Arc expression. If they do, this will have important clinical implications.

Forskolin is already an agonist for PKA. Just omitting it reduces Arc expression over 10-fold.

  1. Bloomer, W.A.; VanDongen, H.M.; VanDongen, A.M. Arc/Arg3.1 translation is controlled by convergent N-methyl-D-aspartate and Gs-coupled receptor signaling pathways. J Biol Chem 2008, 283, 582-592.
  2. Bloomer, W.A.; VanDongen, H.M.; VanDongen, A.M. Activity-regulated cytoskeleton-associated protein Arc/Arg3.1 binds to spectrin and associates with nuclear promyelocytic leukemia (PML) bodies. Brain Res 2007, 1153, 20-33.
  3. Wee, C.L.; Teo, S.; Oey, N.E.; Wright, G.D.; VanDongen, H.M.; VanDongen, A.M. Nuclear Arc Interacts with the Histone Acetyltransferase Tip60 to Modify H4K12 Acetylation. eNeuro 2014, 1.
  4. Oey, N.E.; Leung, H.W.; Ezhilarasan, R.; Zhou, L.; Beuerman, R.W.; VanDongen, H.M.; VanDongen, A.M. A Neuronal Activity-Dependent Dual Function Chromatin-Modifying Complex Regulates Arc Expression. eNeuro 2015, 2.
  5. Hardingham, G.E.; Bading, H. The Yin and Yang of NMDA receptor signalling. Trends Neurosci 2003, 26, 81-89.
  6. Vanhoutte, P.; Bading, H. Opposing roles of synaptic and extrasynaptic NMDA receptors in neuronal calcium signalling and BDNF gene regulation. Curr Opin Neurobiol 2003, 13, 366-371.
